# Trnp1 organizes diverse nuclear membrane-less compartments in neural stem cells

Miriam Esgleas[1,2], Sven Falk[1,2] (iD), Ignasi Forné[3], Marc Thiry[4], Sonia Najas[1,2], Sirui Zhang[5] (iD), Aina Mas-Sanchez[1,2], Arie Geerlof[6], Dierk Niessing[7,8,†], Zefeng Wang[5], Axel Imhof[3,9] (iD) & Magdalena Götz[1,2,9,*] (iD)

## Abstract

TMF1-regulated nuclear protein 1 (Trnp1) has been shown to exert potent roles in neural development affecting neural stem cell self-renewal and brain folding, but its molecular function in the nucleus is still unknown. Here, we show that Trnp1 is a low complexity protein with the capacity to phase separate. Trnp1 interacts with factors located in several nuclear membrane-less organelles, the nucleolus, nuclear speckles, and condensed chromatin. Importantly, Trnp1 co-regulates the architecture and function of these nuclear compartments *in vitro* and in the developing brain *in vivo*. Deletion of a highly conserved region in the N-terminal intrinsic disordered region abolishes the capacity of Trnp1 to regulate nucleoli and heterochromatin size, proliferation, and M-phase length; decreases the capacity to phase separate; and abrogates most of Trnp1 protein interactions. Thus, we identified Trnp1 as a novel regulator of several nuclear membrane-less compartments, a function important to maintain cells in a self-renewing proliferative state.

**Keywords** heterochromatin; mitosis; nuclear speckles; nucleolus phase-transition

**Subject Categories** Chromatin, Transcription & Genomics; Neuroscience; Stem Cells & Regenerative Medicine

The EMBO Journal (2020) 39: e103373

## Introduction

Trnp1 was first identified in breast cancer cells, as a target of TMF1, an E3 ubiquitin ligase, promoting proliferation (Volpe *et al*, 2006) and in a screen for factors regulating neural stem cell (NSC) self-renewal (Pinto *et al*, 2008). Trnp1 is highest expressed in subsets of NSCs that self-renew (Stahl *et al*, 2013) and its overexpression (OE) maintain NSCs in a self-renewal state (Stahl *et al*, 2013; Martinez-Martinez *et al*, 2016). Conversely, Trnp1 knock-down (KD) promotes the differentiation of NSC toward other progenitors, thereby generating folds in the normally smooth mouse brain (Stahl *et al*, 2013). Besides its highly specific expression in neurogenic sites in the nervous system (Stahl *et al*, 2013), Trnp1 is also expressed in many other stem and progenitor cells during development (Metzger, 2011), suggesting a common role in promoting proliferation across tissues. However, how Trnp1 acts at the molecular level in the nucleus is entirely unknown.

The nucleus contains several membrane-less organelles (MLOs) that are generated and regulated through a process of liquid–liquid-phase separation, LLPS (Shin & Brangwynne, 2017). Proteins promoting LLPS share common motifs, including long stretches of intrinsically disordered regions (IDRs) with low amino acid sequence complexity (Uversky, 2018). They can initiate low-affinity transient interaction with several proteins and form homo-oligomers at the same time, acting as multivalent proteins capable to bind transiently to hundreds of proteins (Li *et al*, 2012). These local protein concentrations have four main functions (Sawyer *et al*, 2019). First, MLOs may concentrate and, secondly, exclude components to facilitate and regulate biochemical reactions within the cell. Thirdly, MLOs can sequester key effector molecules away from their site of action. And finally, MLOs may act as reservoirs that permit rapid recycling of molecules to their nearby sites of activity.

Many nuclear MLOs are disassembled during M-phase (Rai *et al*, 2018). However, once reassembled, MLOs can remain stable for long periods of time, even while their components are in a constant state of dynamic flux with the surrounding nucleoplasm (Phair &

1 Physiological Genomics, Biomedical Center (BMC), Ludwig-Maximilians Universitaet Muenchen, Planegg/Munich, Germany
2 Institute for Stem Cell Research, Helmholtz Zentrum Muenchen, German Research Center for Environmental Health, Neuherberg, Germany
3 Protein Analysis Unit, BioMedical Center (BMC), Ludwig-Maximilians-Universitaet Muenchen, Planegg/Munich, Germany
4 Cell and Tissue Biology Unit, GIGA-Neurosciences, University of Liege, C.H.U. Sart Tilman, Liege, Belgium
5 CAS Key Laboratory of Computational Biology, CAS-MPG Partner Institute for Computational Biology, University of Chinese Academy of Sciences, Chinese Academy of Sciences, Shanghai, China
6 Institute of Structural Biology, Helmholtz Zentrum Muenchen, Neuherberg, Germany
7 Group Intracellular Transport and RNA Biology at the Institute of Structural Biology, Helmholtz Zentrum Muenchen, Neuherberg, Germany
8 Department of Cell Biology, BioMedical Center (BMC), Ludwig-Maximilians-Universitaet Muenchen, Planegg/Munich, Germany
9 SYNERGY, Excellence Cluster of Systems Neurology, BioMedical Center (BMC), Ludwig-Maximilians-Universitaet Muenchen, Planegg/Munich, Germany
*Corresponding author. Tel: +49-089218075252; E-mail: Magdalena.goetz@helmholtz-muenchen.de
†Present address: Institute of Pharmaceutical Biotechnology, Ulm University, Ulm, Germany

Misteli, 2000). Examples of nuclear MLOs include nucleoli, which are sites of rRNA biogenesis (Feric *et al*, 2016), Cajal bodies, which serve as an assembly site for small nuclear ribonucleoproteins (RNPs) (Strzelecka *et al*, 2010), nuclear speckles, which are storage compartments for mRNA splicing factors (Spector & Lamond, 2011), heterochromatin (Larson *et al*, 2017; Larson & Narlikar, 2018; Liu *et al*, 2020), superenhancers (Hnisz *et al*, 2017; Sabari *et al*, 2018), transcription complexes (Boija *et al*, 2018; Guo *et al*, 2019; Shrinivas *et al*, 2019), and DNA repair compartments (Kilic *et al*, 2019; Pessina *et al*, 2019). Thus, the nucleus contains several MLOs with crucial functions for the cell. However, so far, most key players for MLO biogenesis and architecture act in one specific compartment and still little is known about how different MLOs are coordinated (Berchtold *et al*, 2018; Rai *et al*, 2018). Here, we show that Trnp1 is a novel low complexity (LC) protein with the capacity to phase separate and co-regulate several nuclear MLOs.

# Results

## Trnp1 is a novel short-lived low complexity protein with the capacity to phase separate

In order to gain insights into the molecular role of Trnp1, we investigated its biochemical properties. Analysis of the primary structure of Trnp1 shows that the most conserved parts (100% homology among all orthologs) are a stretch of 16 amino acids (aa) at its N-terminus and a long alpha helix region (aa87–144) in the central part (Fig 1A) with high probability (80%) to form coiled coils (CC) (Fig 1B). Software tools for disorder prediction IUPred (Dosztanyi *et al*, 2005a) and DisoPred (Dosztanyi *et al*, 2005b) show two regions of low structural complexity: one at the N-terminus (aa1–103) and the other at the C-terminus (aa165–178 and aa196–223) with 82, and 81% of disorder-promoting aa, respectively (Fig 1A), including Trnp1 in the category of LC proteins. Consistent with the behavior of LC proteins, Trnp1-immunostaining shows a punctate pattern, suggestive of condensates in the nucleoplasm (Figs 1C and 5A).

As many IDR proteins self-interact, we co-expressed Trnp1 fused to GFP (Trnp1-GFP, referred to as Trnp1-fusion) and untagged Trnp1 protein in P19 cells that lack endogenous Trnp1 (Fig 1C) and performed co-immunoprecipitation (IP) using α-GFP antibodies followed by Western blot (WB) using α-Trnp1 antibodies. This showed the Trnp1-fusion protein interacting with Trnp1 (Fig 1D) indicating that Trnp1 is able to self-interact. As most LC proteins with the capacity to self-interact also phase separate (Alberti *et al*, 2019), we produced the recombinant Trnp1 protein and used phase-contrast microscopy to examine the formation of phase-separated protein-rich condensates, here referred to as droplets. As Trnp1 is a highly insoluble protein, all the recombinant proteins used in this work were fused to Maltose Binding Protein (MBP) to solubilize them and MBP was cleaved before examining their capacity to form droplets. Notably, Trnp1 recombinant protein, but not the recombinant YFP protein used here as negative control, self-assembles to form droplets in a dose-dependent manner (from 25 to 6.25 μM) under physiological conditions (Fig 1E). Notably, addition of dextran (used here as a crowding agent) caused formation of fibrillary solid aggregates, while RNA addition had very little effects on Trnp1 LLPS and aggregation (Fig 1E). These results indicate that

Trnp1 drives phase separation and hence may play a role in the formation of MLOs.

As MLOs are dynamic structures during the cell cycle, regulated by several processes like dilution effects upon nuclear envelope breakdown in mitosis (Rai *et al*, 2018) or proteolysis (Wang & Zhang, 2019), we analyzed Trnp1 protein stability in P19 cells. When cells were treated with cycloheximide (CHX) to block translation, transfected Trnp1 was degraded within 4 h and hence has a short half-life of less than 2 h (Fig EV1A). This degradation was blocked in the presence of the proteasome inhibitor MG132 (Fig EV1A), indicating that Trnp1 is efficiently and fast degraded via proteasome.

## Structure–function analysis of Trnp1

To understand which parts of the Trnp1 protein were important for its function and self-interaction capacity, we generated deletion constructs (Fig 2A), removing the first conserved 16 aa (Δ1–16) at the N-terminus, extending this deletion to aa 56 to eliminate the proline-rich region (Δ1–56), to aa 97 to delete the entire N-terminal IDR (Δ1–97), or to aa 140 to also remove the predicted alpha helix region (Δ1–140). Likewise, we also deleted the C-terminal IDR including the predicted alpha helix region (Δ95–223) or retaining the alpha helix and only deleting the C-terminal IDR (Δ140–223). As the central alpha helix exhibits a high probability to form CC (Fig 1B), its aa sequence was mutated such as to interfere with the formation of the CC but not of the alpha helix structure (mutCC) (Figs 2A and EV1B). Successful mutation of the CC domain was confirmed using recombinant peptides containing only the alpha helix region (aa 95–161), WT or mutated, and size-exclusion chromatography coupled to an inline static light scattering (SLS) analyzer (Fig EV1C).

We then tested the localization of the different Trnp1 deletions by expressing them together with GFP in cells from embryonic day (E) 14 cerebral cortex (Fig 2B), comprising stem and progenitor cells as well as young neurons, and in P19 cells (Fig EV1D). Notably, all Trnp1 proteins lacking the C-terminal part were detected also in the cytoplasm of both cell types (Figs 2B and EV1D), indicating that this part is responsible for the restricted nuclear localization. While no NLS consensus sequence could be predicted in Trnp1, its C terminus is rich in arginines (Fig 1A), known to be important for nuclear import (Palmeri & Malim, 1999). Conversely, all N-terminal truncated proteins were mostly detected in the nucleus (Figs 2B and EV1D). For accurate quantification of the localization of these deletion constructs, nuclear and cytoplasmic fractionations were performed with P19 cells transfected with these mutant constructs (Fig 2C). While most of the WT Trnp1 and the N terminal truncated proteins are nuclear, the C-terminal deletion constructs are mostly contained in the cytoplasmic fraction (Fig 2C), confirming the crucial role of the C-terminus in nuclear retention of Trnp1. In contrast, the mutCCTrnp1 protein is located in both fractions when expressed in P19 cells (Fig 2C). Notably, its nuclear retention is larger in cells with endogenous Trnp1 (Fig EV1E) and its knockdown (KD) using a previously verified shRNA against the 3′ UTR of endogenous Trnp1 (Stahl *et al*, 2013) reduces the remnant nuclear localization of the mutCCTrnp1 lacking the 3′ UTR region and hence not targeted by the shRNA (Fig EV1F). Thus, the CC domain of

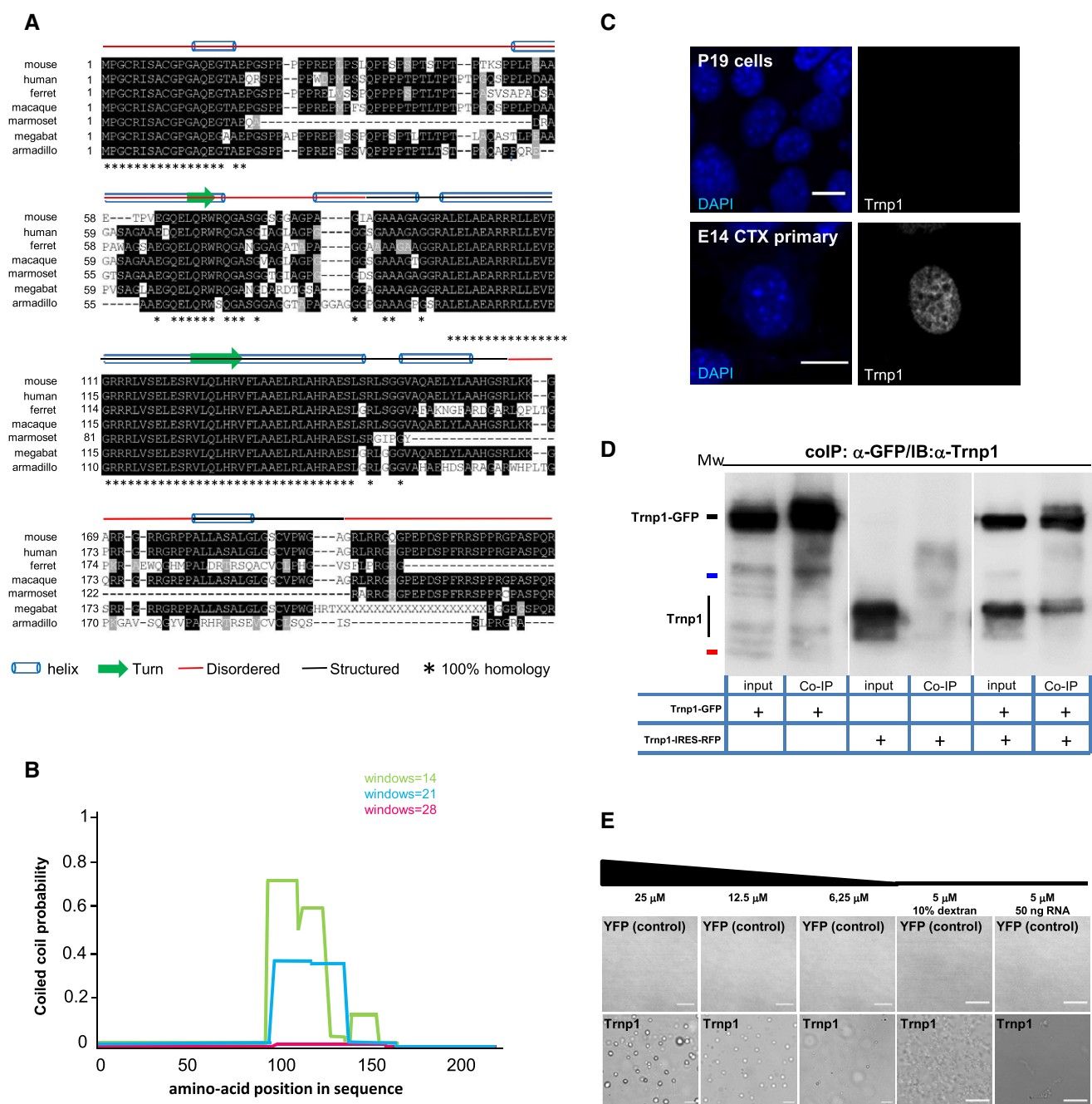

**Figure 1. Characterization of Trnp1 protein.**

A   Clustal alignment analysis of Trnp1 orthologs. Highly conserved and similar residues are shown in black and gray, respectively. Asterisks show 100% conserved residues among all orthologs. Predicted disordered and structured regions are depicted with a red line and a black line, respectively. Cylinders represent alpha-helices, and arrows represent turns.

B   Analysis of the coiled-coil structure of Trnp1 using the prediction tool COILS.

C   Immunostaining of endogenous Trnp1 and DAPI in P19 cells and E14 cortical cells at 1 day *in vitro* (div).

D   Co-IP with anti-GFP and WB with anti-Trnp1 antibodies in lysates from P19 cells transfected with plasmids expressing Trnp1-GFP fusion protein and/or Trnp1-IRES-RFP for 24 h. Mw (molecular weights): 55 kDa (black line); 35 kDa (blue line); 25 kDa (red line). Input: 0.1%; co-IP: all immunoprecipitated proteins.

E   Representative phase-contrast images of *in vitro* phase separation of the recombinant proteins YFP (upper row) or Trnp1 (lower row) at the indicated concentration in 50 mM Sorenson's buffer (pH 7.6) containing 150 mM salt and 2 mM DTT with the crowding agent dextran or RNA when depicted.

Data information: Scale bars: 10 μm (C) and 50 μm (E). In Dextran and RNA condition 10 μm (E).

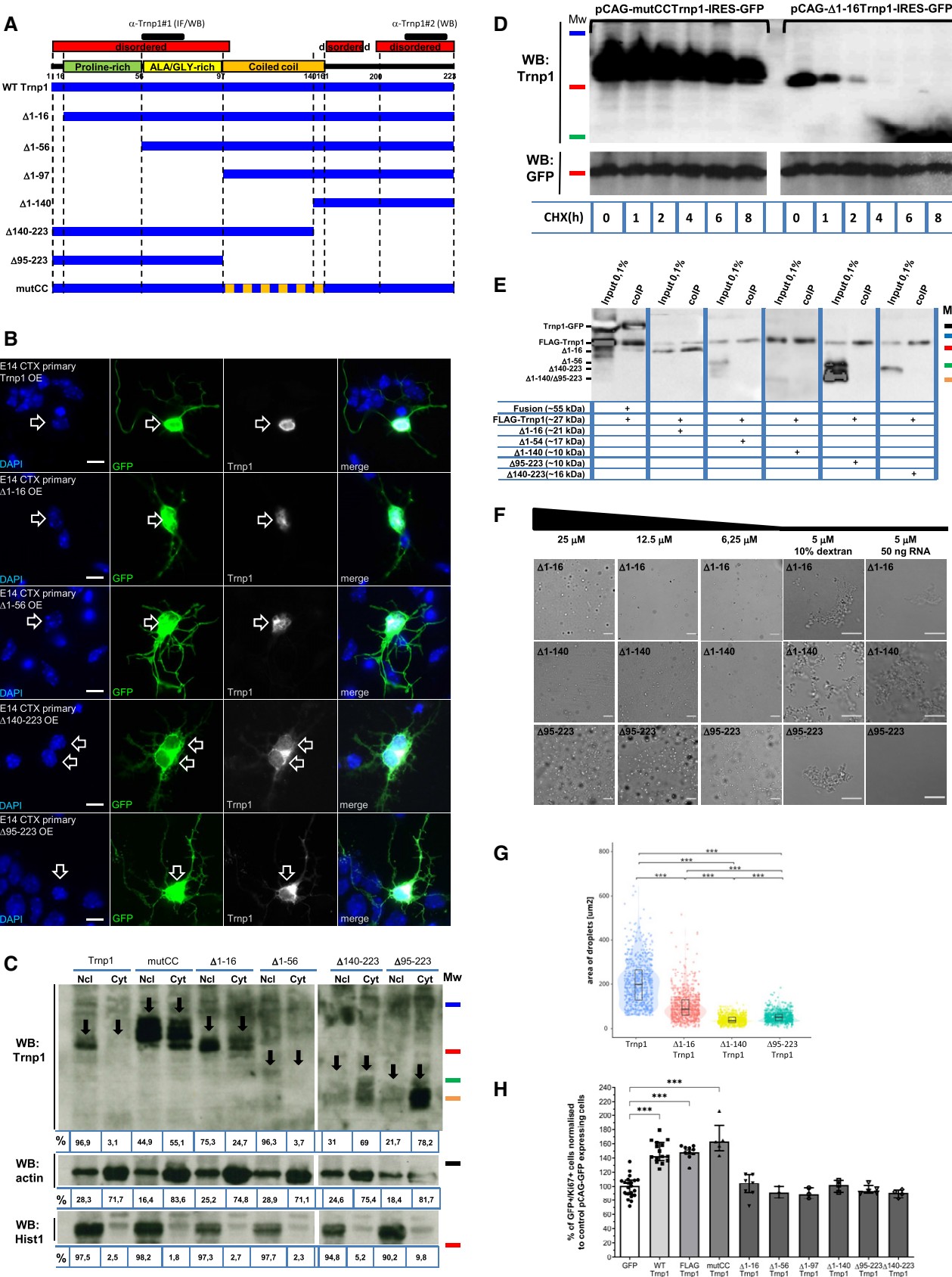

**Figure 2.**

**Figure 2.  Structure–function analysis of Trnp1.**

A  Scheme showing the different Trnp1 truncated/mutant constructs generated here and the region used for generating the Trnp1 antibodies. Note that antibody#1 works for immunofluorescence (IF) and WB while antibody#2 only works for WB.

B  GFP and Trnp1 immunostaining with DAPI labeling in E14 cortical cells transfected for 24 h with plasmids expressing the different truncated/mutant forms of Trnp1 and GFP showing Trnp1 localization (indicated by arrows).

C  Representative WB of cytoplasmic and nuclear-enriched fractions of P19 cells transfected with the different truncated/mutant forms of Trnp1 for 24 h. Hist1 and actin serve as loading controls for nuclear and cytoplasmic fractions, respectively. The values under the anti-Trnp1 blots indicate the percentage of the signal relative to the control set to 100% after normalization by α-actin (ImageJ quantification). The values under the anti-actin and anti-Hist1 blots indicate the percentage of the signal considering that nuclear and cytoplasmic fractions together are 100%.

D  Representative WB showing mutCC Trnp1 and Δ1–16 Trnp1 protein turnover in P19 cells 24 h after transfection in the presence of cycloheximide (CHX) for the time indicated below in hours (h). GFP (half live:24 h) was used as a loading control.

E  Co-IP with anti-FLAG magnetic beads and WB with anti-Trnp1 antibodies showing that FLAGTrnp1 interacts with the GFP-fusion and the Δ1–16Trnp1 proteins but not with other truncated Trnp1 proteins indicating IDRs are crucial for Trnp1 homo-oligomerization. Samples were collected 24 h after transfection of P19 cells with plasmids expressing the indicated proteins depicted in the panel.

F  Representative phase-contrast images of *in vitro* phase separation of the recombinant Δ1–16Trnp1, Δ1–140Trnp1, and Δ95–223Trnp1 proteins at the indicated concentrations in 50 mM Sorenson's buffer (pH 7.6) containing 150 mM salt and 2 mM DTT plus Dextran or RNA when indicated.

G  Violin Dot Plots illustrating quantification of the area of single droplets. Each dot represents a droplet (*N* = 700) formed by the recombinant Trnp1, Δ1–16Trnp1, Δ1–140Trnp1, Δ95–223Trnp1 proteins at 25 μM concentration in a representative experiment of three repetitions under the conditions shown in (F).

H  Histogram depicting the percent of proliferating (Ki67+) cells transfected for 24 h with plasmids expressing control (GFP) or Trnp1-IRES-GFP or its different mutant/truncated forms-IRES-GFP in dissociated E14 cortical cells. Each dot in each condition represents an independent experiment. Biological replicates: GFP = 21; Trnp1 = 15; FLAGTrnp1 = 10; mutCC = 5; Δ1–16 = 7; Δ1–1–56, Δ1–97, Δ1–140 = 3, Δ95–223 = 5, and Δ140–223 = 4.

Data information: Kruskal–Wallis test with Dunn's *post hoc* test (G, H). **$P < 0.01$, ***$P < 0.005$. Mw: 55 kDa (black line); 35 kDa (blue line); 25 kDa (red line), 15 kDa (green line), and 10 kDa (orange line). All experiments were repeated three independent times. Scale bars: 10 μm (B) and 50 μm (F) except for dextran and RNA condition that is 10 μm (F).

---

Trnp1 is important to bind other CC containing proteins and thereby contributes to retain Trnp1 in the nucleus, but it is not required for self-interaction. In addition, the CC is necessary for the rapid turnover of the protein (Fig 2D).

IDRs have been described as important regions for protein self-assembly. As obvious candidates to determine the Trnp1 regions that mediate homotypic interactions, truncated Trnp1 proteins lacking the IDRs (Fig 2A) were tested for their capacity to interact with a FLAG-tagged Trnp1 protein in a co-IP assay using anti-FLAG antibodies. Deletion of either the N- or the C-terminal IDRs abrogates their interaction with the FLAGTrnp1 protein (Fig 2E) demonstrating that both IDRs are important for Trnp1 homotypic interactions. Notably, however, deletion of the first N-terminal 16 aa (Δ1–16) still allowed binding to WT Trnp1. To test the behavior of these deletion constructs in phase separation, we produced the recombinant MBP-Δ1–16, MBP-Δ1–140 (C-term IDR), and MBP-Δ95–223 (N-term IDR) Trnp1 proteins and tested their LLPS capacity (Fig 2F). After MBP cleavage, using the same conditions as for WT Trnp1 (Fig 1E), the size of droplets formed by the truncated proteins was notably reduced (Fig 2F) compared to the full-length WT Trnp1 (Fig 1E). When 25 μM of protein was used, droplet size was reduced by 51.2, 79.8, and 73.8% for the −Δ1–16, −Δ1–140, and −Δ95–223 proteins, respectively, compared to WT Trnp1 (Fig 2G). Thus, the Δ1–16 Trnp1 is strongly impaired in its capacity to phase separate (Fig 2G), but can still interact with WT Trnp1 (Fig 2E) indicating that these properties are not identical. For all the other constructs, however, both properties were impaired. Interestingly, the C-terminal IDR (−Δ1–140) seems to be affected by RNA addition forming fibrils in this condition, while WT and the other deletion constructs do not (Fig 2F). All tested constructs still form fibrils upon addition of the crowding agent Dextran (Fig 2F). Taken together, these results show that both, the N- and C-terminal IDRs, are crucial for the phase separation properties of Trnp1, with deletion of only the first 16 aa of the N-terminal IDR already showing a strong reduction in the LLPS capacity of Trnp1.

### The most conserved part of the N-terminal IDR of Trnp1 is required for its function in promoting proliferation *in vitro* and *in vivo*

Given the pronounced defects in localization and phase separation observed with the different truncation constructs, we examined their effects in one of the main functions of Trnp1, namely to promote proliferation and NSC self-renewal (Stahl *et al*, 2013). We therefore stained E14 cortical cells for the proliferation marker Ki67 after transfection with the different Trnp1 constructs described above (Fig 2A). While Trnp1, tagged or not with FLAG, significantly increased the proportion of proliferating (Ki67+) GFP+ (i.e., transfected) cells, compared to the cells expressing GFP only, already removal of the first 16 N-term aa (Δ1–16) resulted in complete abrogation of this effect (Fig 2H). All larger deletions and the constructs localizing predominantly in the cytoplasm were also no longer promoting proliferation (Fig 2H). Conversely, the mutCCTrnp1 was still able to significantly increase proliferation to a similar extent as the WT Trnp1 (Fig 2H), indicating that the CC is not necessary to maintain cells in a proliferating self-renewing state. Taken together, the nuclear localization and the first 16 aa at the N-terminus are key for Trnp1 promoting proliferation.

To test this also *in vivo*, we electroporated GFP only, Trnp1-IRES-GFP, or Δ1–16 Trnp1-IRES-GFP constructs into the cerebral cortex of E13 embryos and analyzed the distribution of GFP+ cells (Fig 3A–C) 2 days later (E15) dividing a cortical column into five equal-sized bins from ventricular to pial surface (Fig 3D). This analysis already gives an indication of changes in cell identity as proliferating cells are located in bins 1–2, comprising the ventricular and subventricular zones (VZ and SVZ). Indeed, upon Trnp1 expression (Fig 3B) the proportion of GFP+ cells in bin 1, where mostly NSCs are located, was almost double compared to the control (Fig 3D). This was not the case, when Δ1–16 Trnp1 was expressed (Fig 3C and D). Accordingly, also the proportion of proliferating (Ki67+) cells (Fig 3E) and of Pax6+ NSCs (Fig 3F) among the GFP+ cells was significantly increased when Trnp1, but not when Δ1–16Trnp1 was

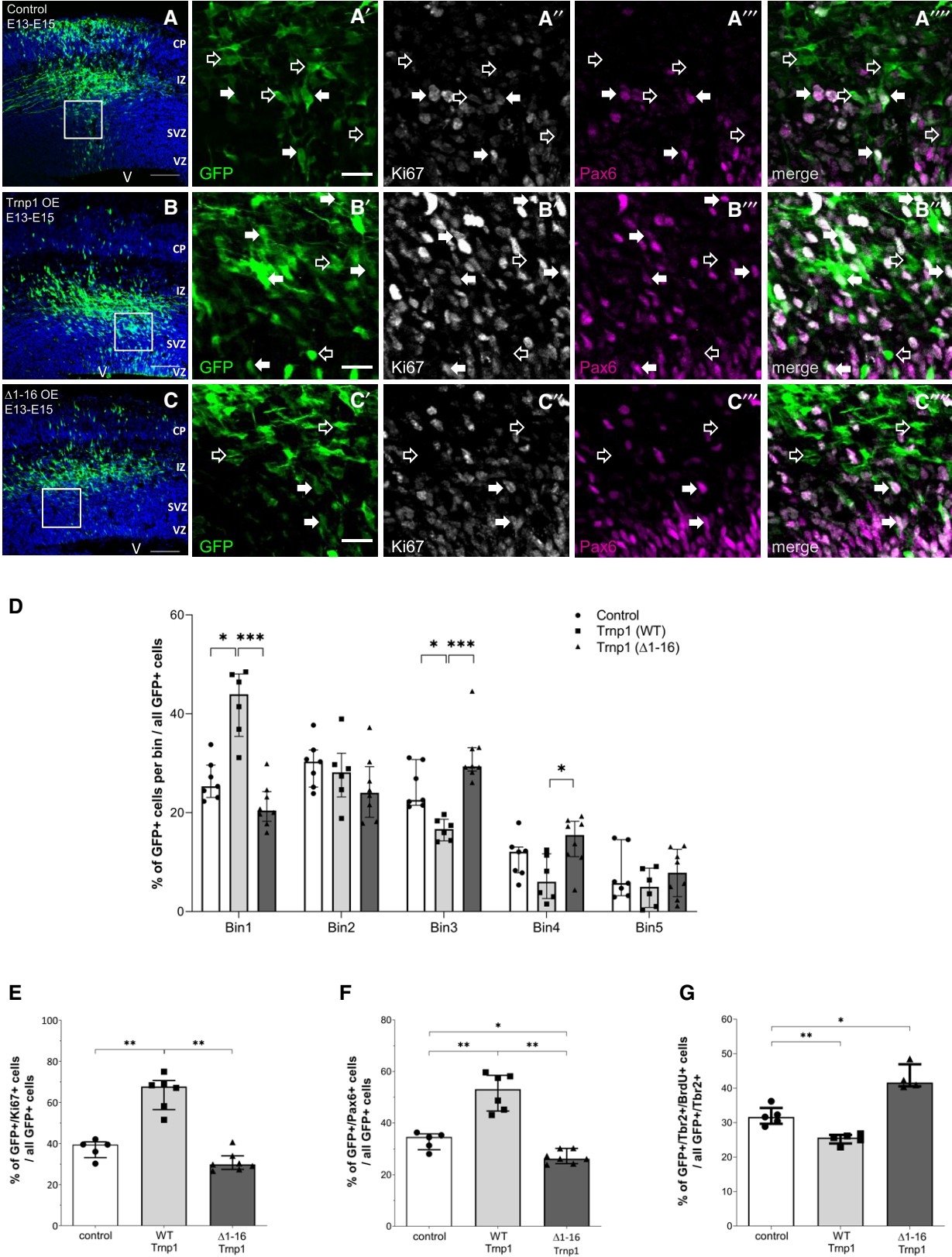

**Figure 3.**

◀

Figure 3. The N-terminal IDR (first 16aa) of Trnp1 is important for promoting proliferation *in vivo*.

A–C  Micrographs showing examples of sections of E15 cerebral cortex 2 days after IUE of the plasmids indicated in (A–C) with magnifications showing the indicated immunostainings. Scale bars: 100 and 20 μm for magnifications. Filled arrows show electroporated cells expressing Pax6 and Ki67 proteins. Open arrows show electroporated cells negative for Pax6 and Ki67 staining. VZ: ventricular zone; SVZ: subventricular zone; IZ: intermediate zone; CP: cortical plate; V: ventricle.
D  Bar and dot plot showing the distribution as percent of GFP+ cells/bin after IUE and staining as in A–C. A cortical column reaching from the ventricular surface (bottom in A–C) to the pial surface (top in A–C) was divided into equal-sized bins with bin 1 at the ventricular surface and bin 5 at the pial surface.
E–G  Bar and dot plots of the proportion of proliferating (Ki67+) cells (E), the number of NSCs (Pax6+), (F) and the proportion of proliferating (BrdU+) TAPs (Tbr2+) (G) as in A–C showing a significant increase in proliferating NSCs upon Trnp1 IUE compared to control or Δ1–16Trnp1 proteins (E, F) and a significant decrease in proliferating TAPs upon Trnp1 IUE and increase upon Δ1–16Trnp1 IUE compared to control (G) at E15, 2 days after IUE.

Data information: Biological replicates: in (D): control = 7; Trnp1 = 6; Δ1–16 = 8, in (E and F): control = 5; Trnp1 = 6; Δ1–16 = 7, in (G), control = 5; Trnp1 = 5; Δ1–16 = 4 embryos from 2 to 4 different litters. Data are presented as median ± IQR. Kruskal–Wallis test with Dunn's *post hoc* test (D) and Mann–Whitney test (E–G). *$P < 0.05$, **$P < 0.01$, ***$P < 0.005$.

electroporated compared to control (Fig 3A and B). Interestingly, Trnp1 affected NSC proliferation specifically, as transit-amplifying progenitors (TAPs, also called basal progenitors) labeled by the transcription factor Tbr2 rather decreased their proliferation. This was measured by the number of cells found in S-phase using a short pulse (1 h) of the DNA-base analog BrdU upon Trnp1 overexpression (Fig 3G). In contrast, Δ1–16 Trnp1 expression had the opposite effect rather increasing TAP proliferation (Fig 3G), while decreasing the proportion of Pax6+ NSCs compared to the control cells (Fig 3F). Thus, also *in vivo*, deletion of the first 16 aa of the N-terminal IDR of Trnp1 not only lost, but partially even antagonized the function of Trnp1 in maintaining proliferation and retaining the NSCs (Pax6+) in a self-renewing state.

### Trnp1 interacts with proteins from nuclear membrane-less compartments

To identify the molecular basis of Trnp1 function, we used mass spectrometry (MS) to examine how and where its potential function in phase separation may be relevant in the nucleus. After ensuring that the addition of the FLAG at the N-terminus of Trnp1 (FLAGTrnp1) did not affect Trnp1 localization (Fig EV2A) and function (Figs 2H and EV2B–E), FLAG-tagged Trnp1 and non-tagged Trnp1 as control were transfected in P19 cells and 24 h later the DNAseI-treated nuclear extracts were immunoprecipitated using anti-FLAG magnetic beads. The eluted proteins were identified and quantified by label-free LC-MS/MS. Non-specific interactors were discriminated by comparison with the proteins identified in the untagged Trnp1 control samples.

We found 351 proteins to be specifically enriched [relative change > 1.9 (the lowest relative change which could be confirmed); *P* value < 0.05] (Dataset EV1). Nine Trnp1 interactors were randomly chosen for immunoprecipitation using specific antibodies and WB using anti-Trnp1 antibody (Fig 4A) confirming the interactions observed by MS. Most of these interactions were not RNA dependent as treating nuclear extracts with Benzonase did not alter their binding to Trnp1 (Fig 4A). Among the Trnp1 interactors, many proteins are associated with nucleoli (36.18%), splicing (9.97%), chromatin organization (4.55%), and cell cycle process (2.56%) (Fig 4B). As deletion of the first 16aa had significantly reduced LLPS and abolished the function of Trnp1, we repeated the above experiment using the FLAGΔ1–16Trnp1 construct and the non-tagged Δ1–16Trnp1 as negative control. Strikingly, interactors from all three hubs were lost with very few significant interactions remaining (Fig 4C). Thus, the highly conserved 1–16aa in the N-terminal IDR

of Trnp1 are crucial for interaction of Trnp1 with proteins of nuclear MLOs in line with their role in LLPS and Trnp1 function.

The functional enrichment tool DAVID (Huang da *et al*, 2009) revealed significantly enriched categories (in biological process, cellular component, and Kyoto Encyclopedia of Genes and Genomes, KEGG, pathway) for all Trnp1-interacting proteins identified by MS (Dataset EV2), such as GO terms associated with rRNA processing and ribosome biogenesis, RNA splicing and mRNA processing, chromatin modification, and transcription (Fig EV2F and Dataset EV2 DAVID Biological process). Trnp1 interactors were enriched for the KEGG pathways "Ribosome biogenesis", "Ribosome", "Spliceosome", "RNA transport", and "mRNA surveillance pathway" suggesting a novel role of Trnp1 in these processes (Dataset EV2 DAVID KEGG pathways). The GO term and KEGG pathway analysis correlated very well with analysis using the Search Tool for the Retrieval of Interacting Genes/Proteins (STRING) 9.0 (http://string-db.org/) limiting this to experimentally validated interactors. STRING analysis clearly shows three main interaction modules: (i) proteins involved in nucleoli and ribosome biogenesis, (ii) chromatin remodeling factors involved in condensation of chromatin, and (iii) RNA splicing proteins (Fig 4D). Thus, Trnp1 interacts predominantly with proteins localized in nuclear MLOs formed by phase separation (Banani *et al*, 2017).

### Trnp1 protein surrounds nucleoli and interacts with components of the granular component

Given the intriguing hubs of proteins interacting with Trnp1 highlighting nuclear MLOs, such as nucleoli, we used super-resolution confocal microscopy to visualize the localization of endogenous Trnp1 and the nucleolar proteins nucleophosmin (B23) and nucleolin (Ncl) (Fig 5A) in E14 cortical cells. Both nucleolar proteins are located in the outer granular component (GC), the most external nucleolar compartment formed by LLPS (Feric *et al*, 2016). Consistent with its phase separation properties, Trnp1 is localized in small condensates throughout the nucleus that are notably increased surrounding the nucleoli overlapping with B23 and Ncl (Fig 5A, for overlap, see line scan). Consistent with this spatial localization, Ncl but not Fbl1 (localized in the dense fibrillary component (DFC)) was among the Trnp1 interactors identified by MS and confirmed by immunoprecipitation for either Flag (Trnp1) or Ncl (Fig 5B). Interestingly, the N-terminal truncated Trnp1 proteins lose their perinucleolar localization and are also found within the nucleoli (Fig EV3A, for overlap, see line scan). Although the localization of the nucleolar protein Ncl was not impaired by these mutants

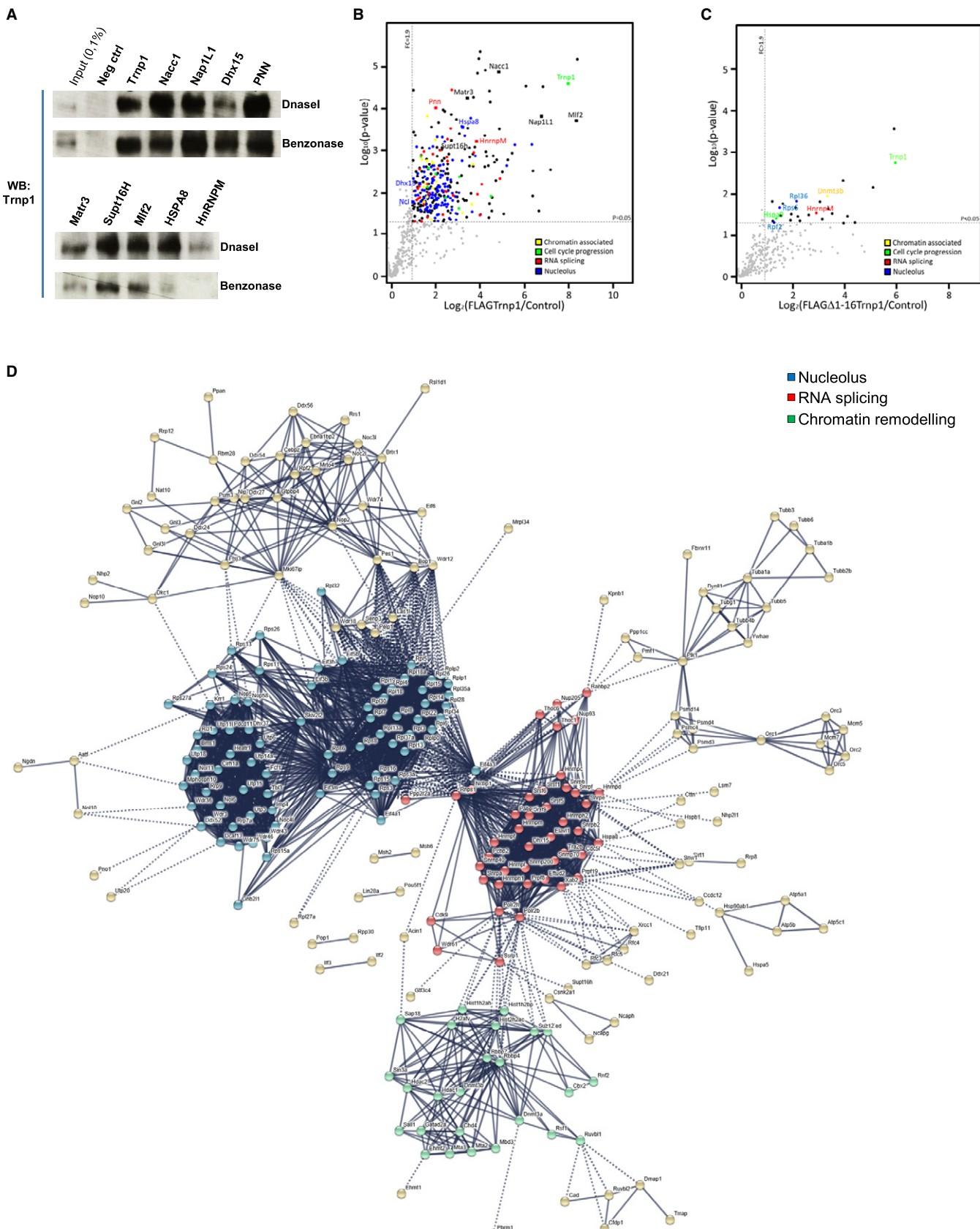

**Figure 4.**

**Figure 4. Trnp1 interacts with proteins involved in ribosomal biogenesis, splicing, and chromatin remodeling.**

A    WB showing co-IP performed with the antibodies indicated on top of the lanes in nuclear lysates treated with the nucleases indicated to the right of the panel. P19 cells were transfected with plasmids expressing FLAGTrnp1 for 24 h and immunoblotted with anti-Trnp1 antibodies. As negative control, no antibody was added to the co-IP.

B, C   Volcano plot showing the mean difference in the protein iBAQ between FLAGTrnp1 (B) and FLAGΔ1–16Trnp1 (C) interacting proteins vs. control plotted against the *P*-value. Proteins used to experimentally confirm the MS results are indicated with squares and the protein name in (B, C).

D    STRING analysis of Trnp1 interactors (B). The disconnected nodules are not shown. Protein cluster has been color coded (green for chromatin association, red for RNA splicing, blue for proteins involved in nucleolus function, and light yellow for the rest).

Data information: In panels (B and C), black dotted lines indicate the fold change (FC > 1.9) and the significance cutoff (*t* test; *P* > 0.05). Each dot represents a protein. Proteins participating in different functions have been color coded (yellow for chromatin association, green for cell cycle, red for RNA splicing, blue for proteins involved in nucleolus function, and black for the rest).

(Fig EV3A), we could observe that Fbl1 and ribosomal protein Rpl26, both necessary for the biogenesis of mature ribosomes (Chen & Huang, 2001; Robledo *et al*, 2008), were mislocalized upon high expression of the Trnp1 N-terminal truncated proteins. While Rpl26 was mostly localized in the periphery rather than inside of nucleoli, Fbl1 was diffuse (Fig EV3B and C), suggesting a role of Trnp1 in the protein traffic between the nucleus and the nucleoli.

Given this localization and specific interaction with GC nucleolar proteins, we were interested to determine the functional consequences of raising Trnp1 levels on nucleoli structure and function.

## Trnp1 regulates nucleoli, chromatin architecture, and nuclear speckles

To explore the effects of Trnp1 on nucleoli, we first used TEM (transmission electron microscopy) of P19 cells transfected with GFP (as control) or Trnp1-IRES-GFP. One day after transfection, GFP+ cells were isolated by FACS and processed for TEM (Fig 5C–E). TEM revealed notable changes in the nucleoli upon Trnp1 expression (Fig 5C). Although the three compartments were present, Trnp1 promoted an enlargement in the nucleolar interstice in 51.4% of Trnp1-expressing cells (120 cells analyzed), while we could not observe this phenotype in any of the control cells (132 cells analyzed) (Fig 5C; red arrows). These nucleolar spaces appear around the two fibrillar components (DFC and FC), partially separating them from the GC. We could also observe that condensed chromatin blocks associated with the nuclear envelope and/or nucleoli appear slightly larger in cells overexpressing Trnp1 compared to control cells (Fig 5D, red arrows). However, these condensed chromatin blocks are not very frequent under both conditions and most can be observed associated with the nuclear envelope and/or nucleoli. Finally, TEM micrographs also show a significant increase in the presence of dense strands within interchromatin granules clusters (also called nuclear speckles) in cells expressing Trnp1 compared to control cells (Fig 5E). While in the control only 8% of cells contain clusters of interchromatin granules with dense cord, this number is increased to 51.4% in Trnp1-expressing cells (Fig 5E; red arrows). Notably, all cells presenting an enlargement in the nucleolar interstice also contain clusters of interchromatin granules with dense cords, suggesting a common role of Trnp1 in both phenotypes. Trnp1 expression does not affect the levels of nucleolar proteins (Fig 5F), indicating that the effects of Trnp1 on the nucleolus are dependent rather on Trnp1's structural function than its potential role in gene expression (Stahl *et al*, 2013) or regulating nucleolar protein levels. The changes in the architecture of condensed chromatin, nuclear speckles, and nucleoli after Trnp1 manipulation are intriguing as all these MLOs are formed by

LLPS and are rich in Trnp1 interactors as observed by MS (Fig 4D). Thus, Trnp1 may play a role in regulating several nuclear MLOs simultaneously.

## Trnp1 regulates the dynamics of nucleoli during cell cycle and M phase length

Given that Trnp1 modifies the heterochromatin associated with the nucleolus and also interacts with many proteins related to nucleolar function, including proteins essential for nucleolus biogenesis such as UBF1 (Grob & McStay, 2014) and Ncl (Ma *et al*, 2007; Ugrinova *et al*, 2007), we examined if Trnp1 may regulate nucleoli dynamics during cell cycle. To visualize if Trnp1 is responsible of changes in nucleoli assembly and disassembly, P19 cells were transfected with plasmids expressing Fbl1-GFP fusion protein (for nucleoli visualization) together with RFP alone, Trnp1-IRES-RFP, or Δ1–16Trnp1-IRES-RFP. Imaging started 22 h after transfection with confocal images taken every 10 min for a period from 16 to 72 h (Fig 6A). As expected during M-phase (Angelier *et al*, 2005), the large GFP-Fbl1 foci disappeared for some time and re-appeared as small foci that then fuse and form again larger structures (Fig 6A). To quantify the effects of Trnp1 on nucleoli, we measured the number, size, and total area of GFP-Fbl1 foci per cell over time starting 180 min before and finishing 180 min after separation of sister chromatids in M-phase (*t* = 0) (Fig 6A–C). While Trnp1 expression did not affect the number of GFP-Fbl1 foci (Fig 6B), it significantly increased their size (Fig 6A) and the total area of GFP-Fbl1 foci per cell (Fig 6C), indicating that Trnp1 expression increases the size of the existing nucleoli, but not the formation of new ones.

We further noted that Trnp1 shortened the period of GFP-Fbl disappearance (Fig 6A and C), while nucleoli re-assembly was not affected (Fig 6C). This prompted us to explore the dynamics of the cell cycle and especially the length of M-phase, that is normally closely coordinated with nucleoli dynamics. To do so, we expressed a H2B-Venus fusion protein to monitor mitosis from the condensation of the chromatin starting in prophase to the de-condensation of chromosomes at the end of telophase (Fig 6D). Interestingly, these experiments showed that Trnp1 expression significantly shortened the duration of M-phase (Trnp1 = 54.3 min) compared to control transfected cells (RFP = 70.8 min) (Fig 6E). Synchronizing P19 cells and measuring the different mitotic phases at different time points by DAPI confirmed the specific effect of Trnp1 on entry into M-phase as Trnp1 transfected cells left interphase and entered prophase significantly faster than the controls (Fig 6F). Mitotic index analysis in the E15 developing cortex 24 h after in utero electroporation (IUE) of plasmids expressing GFP as control, Trnp1-IRES-GFP, or the Δ1–16Trnp1-IRES-GFP corroborates the faster M-phase provoked by

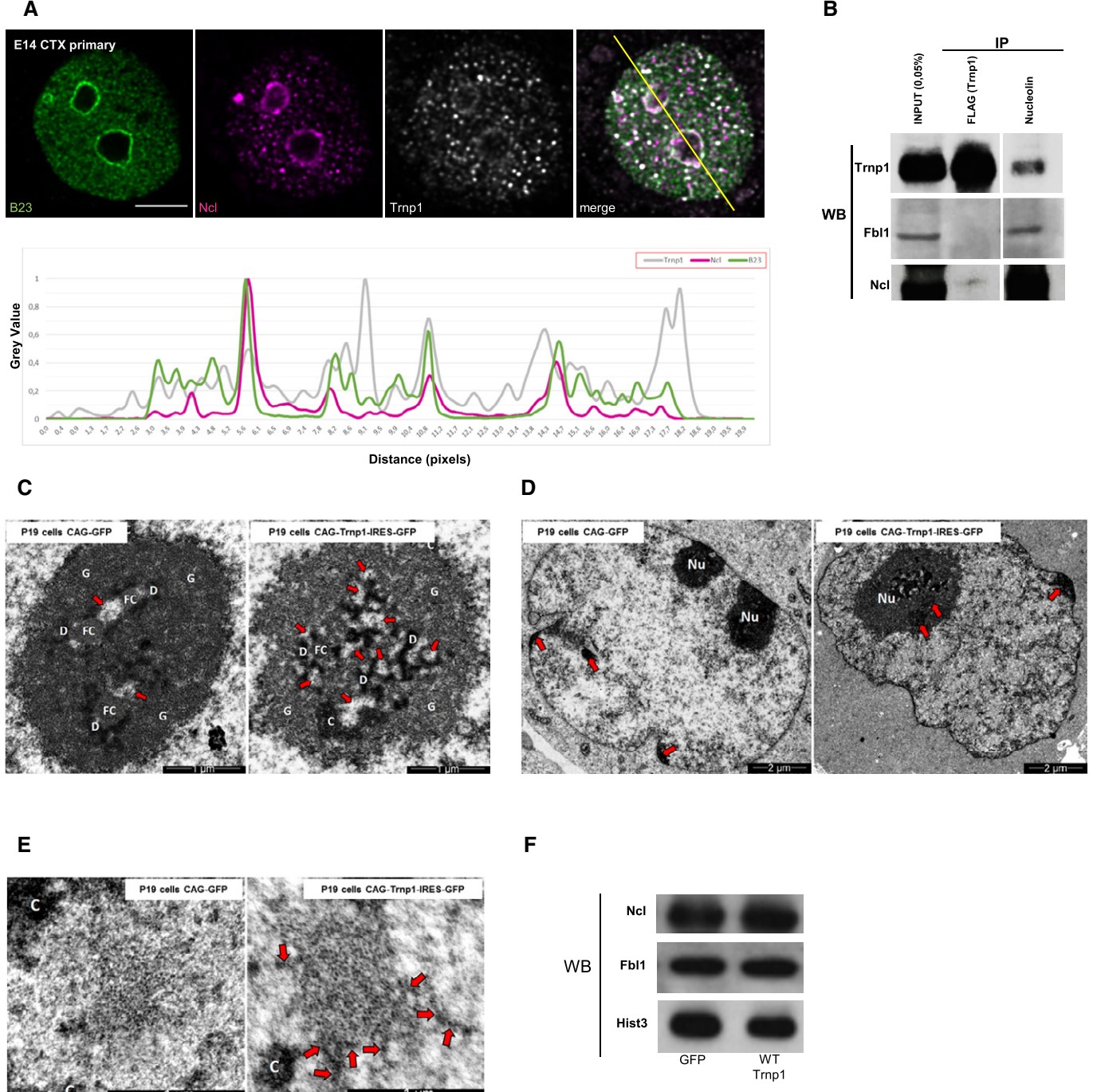

**Figure 5. Trnp1 changes the architecture of nucleolus, condensed chromatin, and nuclear speckles.**

A   Super-resolution confocal image of endogenous nucleophosmin (B23), nucleolin (Ncl), and Trnp1 proteins in E14 cortical cells at 1div. The yellow line indicates the position of the intensity line scan plot (below confocal panels) measuring the gray value of each staining (Trnp1 = gray; Ncl = purple and B23 = green). Scale bars, 5 μm.

B   Co-IP experiments with lysates from P19 cells expressing FLAGTrnp1 for 24 h. IPs and WB with the antibodies indicated show interaction between Trnp1 and Ncl, but not with Fbl1, consistent with the MS data.

C–E   Ultrastructural analysis depicting in the nucleolus (arrows for nucleolar interstice) (C), condensed chromatin blocks (arrows) (D) and nuclear speckles or interchromatin granule clusters (arrows for dense cords) (E) in the nucleus of P19 cells transfected with plasmids expressing Trnp1-IRES-GFP or GFP in control cells FACS sorted and fixed 24 h after transfection. Scale bars, 2 μm (D) and 1 μm (C and E). Nu: nucleolus; G: granular component; D: dense fibrillar component; FC: fibrillar center; C: condensed chromatin.

F   WB of lysates from P19 cells transfeted for 24 h with plasmids expressing GFP only or Trnp1-IRES-GFP with antibodies indicated.

Data information: All experiments were done at least in duplicate.

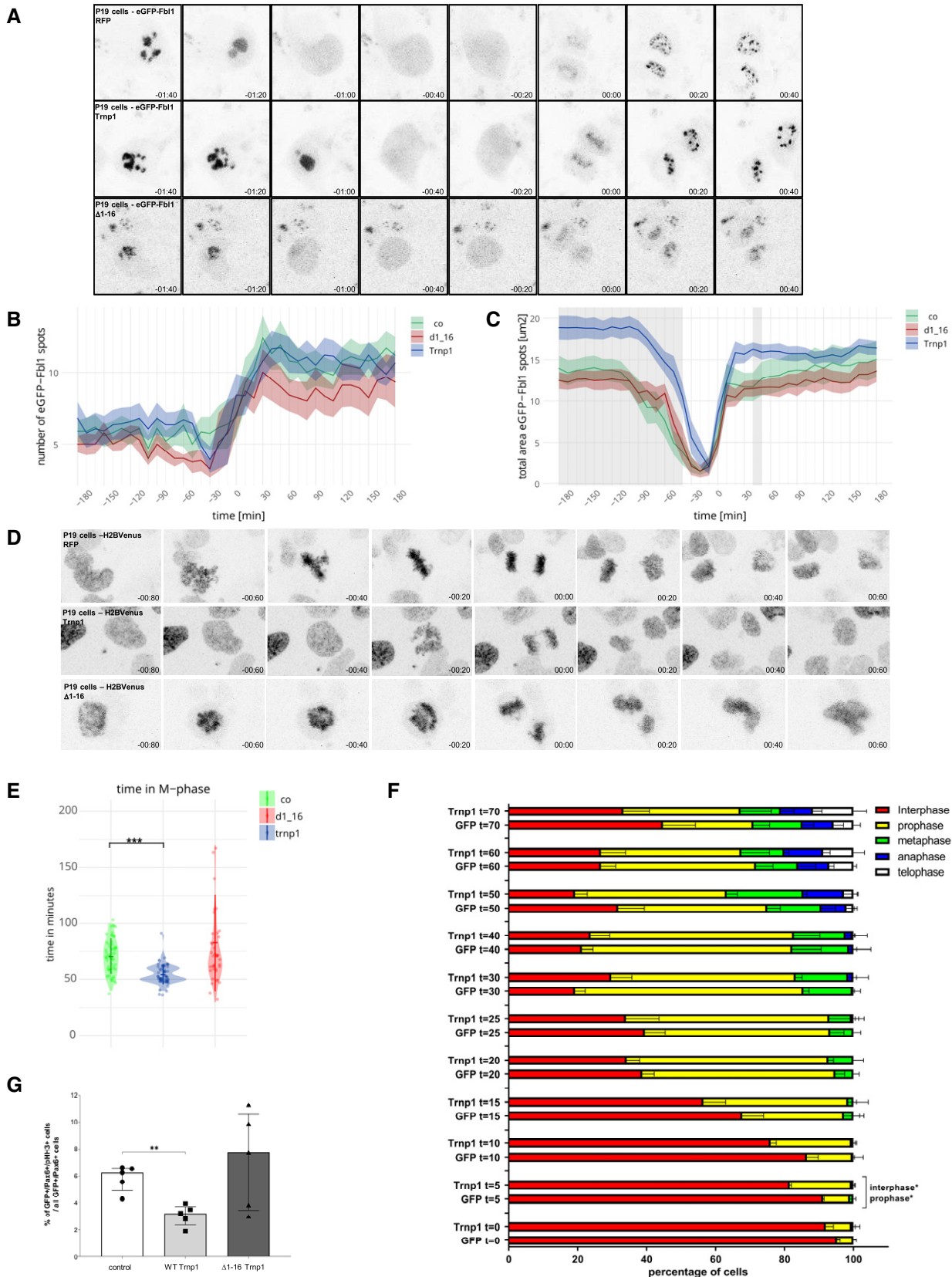

**Figure 6.**

**Figure 6.  Trnp1 modulates nucleoli disassembly and M-phase duration *in vitro*.**

A  Representative time-lapse sequence of GFP-Fbl1 in P19 cells expressing the plasmids indicated defining *t* = 0 when the chromatids separate (anaphase).

B, C  Analysis of the number (B) and the total area (C) of the GFP-Fbl1 foci in P19 cells expressing the constructs indicated for a period of 360 min. Bandwidth represents SEM, and gray areas represent time points with significant differences between the control and the Trnp1-expressing conditions. The Δ1–16Trnp1 expression does not show any statistical difference compared to the control condition.

D  Representative time-lapse sequence including one mitosis of H2B-Venus in P19 cell expressing the indicated constructs.

E  Volcano Plot illustrating the length of M-phase from condensation to de-condensation of chromosomes in minutes. Each dot represents an individual cell from 3 different biological replicates. GFP = 61 cells, Trnp1 = 62 cells, and Δ1–16Trnp1: 40 cells.

F  Bar graph showing the percentage of cells expressing Trnp1-IRES-GFP or GFP as control in each phase of the mitosis (indicated in different color) at different times after release from synchronization in G2. 100 cells per condition and time were analyzed.

G  Bar and dot plots depicting the proportion of GFP+ NSCs (Pax6+) in mitosis (pHH3+) showing a significant decrease 24 h after Trnp1 compared to control IUE. *N* = 4 brains from embryos of 2 to 3 different litters.

Data information: Imaging started 22 h after transfection with confocal images taken every 10 min for a period of 16–72 h. Time 0: chromatid separation. Results are shown as mean ± SEM (B and C) or as median ± IQR (E–G). Mann–Whitney test (B, C, E, F) and Kruskal–Wallis test with Dunn's *post hoc* test (G). *$P < 0.05$; **$P < 0.01$; ***$P < 0.005$.

Trnp1, but not Δ1–16Trnp1 (Fig 6G). Taken together, these results demonstrate a role of Trnp1 in the disassembly of nucleoli, which occurs in prophase, but not in their re-assembly occurring during end of telophase and beginning of G1. This correlated with the faster progression into prophase, suggesting that Trnp1 coordinates nucleolar dynamics and mitosis progression. Consistent with the data shown above, none of these effects were observed in cells transfected with Δ1–16Trnp1 further demonstrating that this small region of the N-terminal IDR is crucial for Trnp1 function regulating M-phase length and affecting nucleoli size and dynamics.

## Trnp1 regulates the size of nucleoli and heterochromatin in the developing cerebral cortex *in vivo*

As the above data were obtained upon overexpression of Trnp1 in P19 cells *in vitro*, a cell type not expressing endogenous Trnp1, we proceeded to examine the novel role of Trnp1 in regulating nuclear MLOs in the developing cerebral cortex *in vivo*. Staining for the nucleolar proteins B23 and Fbl in sections of E15 cortex was notably more prominent in NSCs in the VZ, where endogenous Trnp1 is expressed, compared to the TAPs localized in the SVZ that lack endogenous Trnp1 (Stahl *et al*, 2013; Fig EV4A). To analyze the specific role of Trnp1 on nucleoli, we electroporated plasmids expressing GFP, Trnp1-IRES-GFP, Δ1–16Trnp1-IRES-GFP, and the previously described shRNA against Trnp1 and its specific control, both expressing also GFP (Stahl *et al*, 2013) at E13 and analyzed the nucleoli (using Ncl staining) in GFP+ cells 2 days later (Fig 7A–C, G and H). We focused our analysis on NSCs in the VZ (bipolar cells with the morphology of radial glial cells that were typically Pax6+) and above the VZ, referred to as non-VZ, focusing on multipolar TAPs that were typically Tbr2+ (Fig EV4B–D), excluding the neurons in the cortical plate. First, we measured the size of the nucleus that was not affected by Trnp1 overexpression compared to control cells (Fig 7D). Quantification of nucleoli size and number by Ncl immunostaining in GFP-expressing control cells (Fig 7A) revealed, however, significant differences in both, number (Fig 7E) and size (Fig 7F) of nucleoli in the VZ compared to the non-VZ, correlated with the endogenous levels of Trnp1 (high in VZ, low in non-VZ). Upon Trnp1 overexpression (Fig 7B), we also found a significant increase in nucleoli size (Fig 7F), but not their number (Fig 7E), in the non-VZ where endogenous Trnp1 is low. This is consistent with the effects seen in P19 cells (that do not express endogenous Trnp1) with Trnp1 overexpression increasing size of

nucleoli, but not regulating formation of new nucleoli (Fig 6B and C). Conversely, Δ1–16Trnp1 expression (Fig 7C) significantly reduced the number (Fig 7E) and size (Fig 7F) of nucleoli in the VZ, but not the non-VZ, thereby acting as a dominant-negative form in this context. Importantly, knock-down of Trnp1 (Fig 7G and H) did not affect the number of nucleoli (Fig 7J), but resulted in reduction in nucleolar size in NSCs in the VZ (Fig 7K), corroborating that endogenous Trnp1 in VZ cells increases nucleolar size *in vivo*. Surprisingly, Trnp1 knock-down increased nuclear size (Fig 7I), but as nucleoli size was normalized to the nuclear area, this cannot explain its effects on nucleolar size.

## Trnp1 affects size and function of several nuclear MLOs

Given that Trnp1 interacts with protein hubs of other nuclear MLOs, such as splicing and repressive chromatin complexes, and most of these interactions are lost for the Δ1–16Trnp1 we aimed to determine whether Trnp1 could also be responsible for the modification of the architecture of any of these MLOs *in vivo*. As Trnp1 localizes to euchromatin surrounding the DAPI-rich areas of heterochromatin (Stahl *et al*, 2013; Fig EV4E), we used the same IUE sections described above (Fig 7A–C) and examined the condensed chromatin spots visible in DAPI staining (Fig 8A–C). The number and size of heterochromatin spots in GFP-expressing control cells was similar in VZ and non-VZ (Fig 8D and E). In Trnp1 electroporated cells, we could not observe differences in the number of DAPI-rich spots (Fig 8D). The Δ1–16Trnp1 mutant, however, decreases the number of heterochromatin spots compared to control cells (Fig 8D), indicating that this truncated form of Trnp1 behaves as a dominant negative. As also observed for nucleoli, Trnp1 overexpression promotes a significant increase in the size of heterochromatin spots (Fig 8E), while IUE of the Δ1–16Trnp1 again acts as a dominant negative (Fig 8E). Thus, Trnp1 overexpression in the developing cerebral cortex *in vivo* affects the size of two MLOs simultaneously increasing nucleoli and condensed chromatin spots.

As Trnp1 also affected nuclear speckles when overexpressed in P19 cells, we next examined the role of Trnp1 in regulating alternative splicing in cerebral cortex NSCs *in vivo*, using the above described shRNA constructs (Fig 7G and H). In order to determine the earliest changes not yet affected by the altered cell type composition, we collected GFP+ cells 1 day after IUE with shcontrol or shTrnp1 constructs by fluorescence-activated cell sorting (FACS) (Fig EV4F). RNA sequencing showed a total of 237 differentially

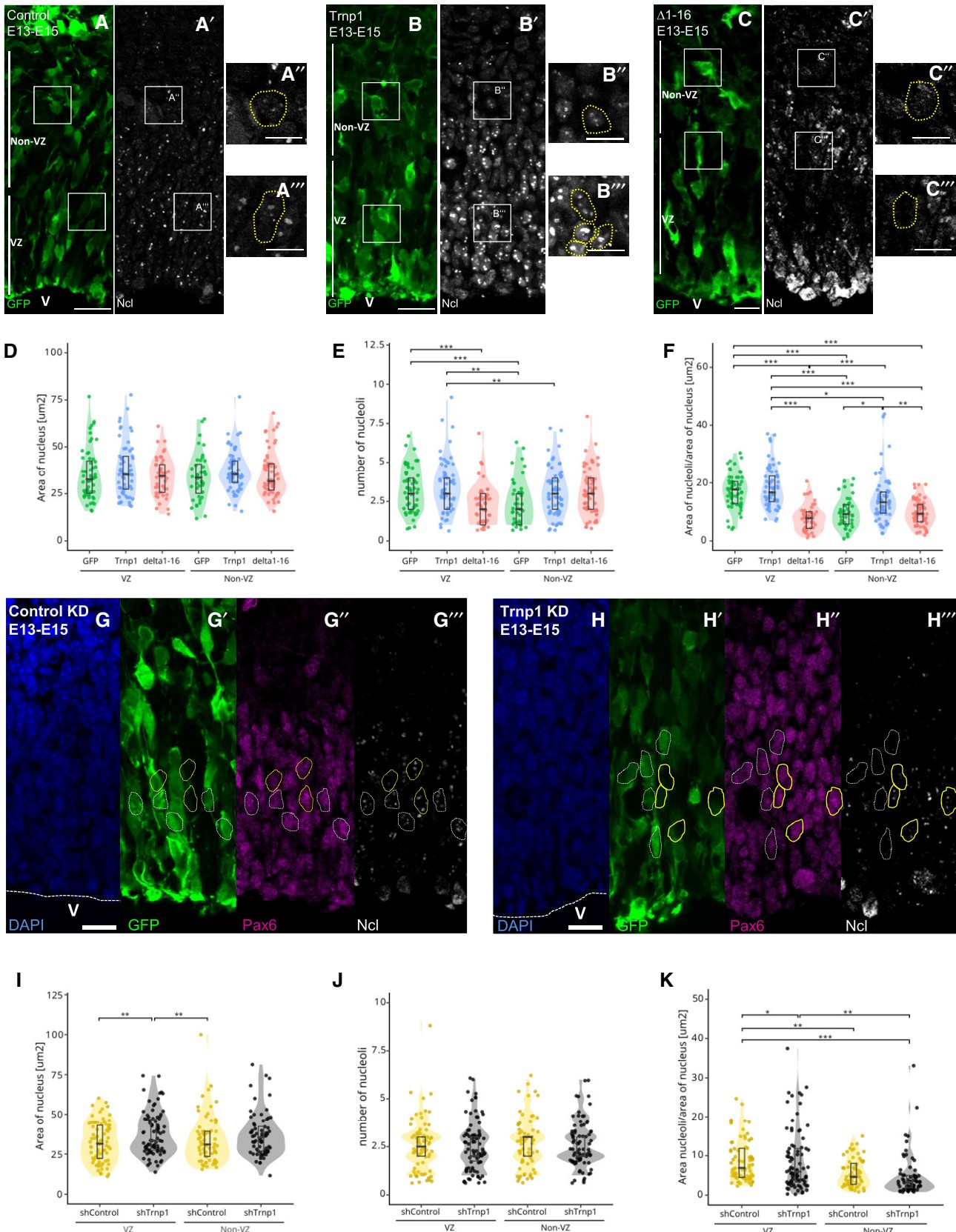

Figure 7.

◀

**Figure 7.  Trnp1 regulates size of nucleoli *in vivo*.**

A–C  Micrographs showing GFP (green: electroporated cells (A–C)) and Ncl (gray: nucleoli (A′–C′)) immunostaining of E15 cerebral cortex 2 days after IUE with plasmids indicated. Position of magnifications shown on the right side is depicted by a white square. V = ventricle; VZ: ventricular zone. Yellow dotted shapes show examples of nucleoli in nuclei of electroporated cells from the VZ and the non-VZ. Scale bars: 20 and 10 µm for magnifications.

D–F  Violin plots depicting the nuclear size (D) nucleoli number (E) and nucleoli size (F) from cells electroporated as in A–C with the constructs and the areas indicated on the *x*-axis. Note that in Trnp1, but not Δ1–16Trnp1, overexpression is sufficient to significantly increase nucleolar size in the non-VZ (where most cells lack endogenous Trnp1). $N = 10$ cells per region from 6 different control and Trnp1-IRES-GFP electroporated embryos and $N = 15$ per region from 3 to 4 different Δ1–16Trnp1-IRES-GFP electroporated embryos from at least 4 litters.

G, H  Micrographs showing GFP (green: electroporated cells), Pax6 (magenta for NSCs), and Ncl (gray for nucleoli) immunostainings in sections of E15 cerebral cortex 2 days after IUE with plasmids indicated. Scale bar: 20 µm. Yellow dotted shapes show examples of nuclei in electroporated, and yellow nondotted shapes show examples of nuclei non-electroporated cells from the VZ and the non-VZ.

I–K  Violin plots depicting the nuclear size (I), nucleoli number (J) and nucleoli size (K) from cells electroporated as in (G, H) with the constructs and the areas indicated on the *x*-axis. Note that Trnp1 knockdown significantly decreases nucleolar size in the VZ (where most cells express endogenous Trnp1). N = 10 cells per region from 7 embryos per condition from at least 4 litters.

Data information: Each dot represents a cell. Boxes in violin plots represent median ± IQR. Kruskal–Wallis test with Dunn's *post hoc* test. *$P < 0.05$, **$P < 0.01$, ***$P < 0.005$.

regulated splicing events and a bias toward a decrease in exon skipping events (Fig 8F), corroborating an influence of Trnp1 levels on alternative splicing.

Given these functional changes along with the structural changes we observed above, we were keen to determine whether function of nucleoli was also affected. Toward this end, we used the OPP-puro click assay to determine the translation rate in P19 cells by quantifying cellular fluorescence using microscopy (Fig 8G) or FACS (Fig 8H). Indeed, Trnp1 expression significantly increased translation (Fig 8G and H) and we could show that this was not due to an increase in ribosomal RNA transcription that was not affected by Trnp1 (Fig 8I). Notably, the expression of Δ1–16Trnp1 also shows an increase in the rate of translation compared to the control cells, but not as high as the one produced by the expression of the WT Trnp1 protein (Fig 8G and H), indicating that the first 16 amino acids are also relevant for this function. This result together with MS analysis showing interaction with many proteins involved in ribosome biogenesis suggests that Trnp1 probably regulates translation by regulating the biogenesis of new ribosomes in the nucleoli. Taken together, Trnp1 levels not only regulate size and architecture, but also function of several nuclear MLOs.

## Discussion

Here, we uncovered the molecular function of Trnp1 interacting with protein hubs associated with several nuclear MLOs, nucleoli, nuclear speckles, and condensed chromatin, mostly via its N-terminal IDR. We found that Trnp1 regulates the size, architecture, and function of these MLOs and is thus the first nuclear protein regulating these MLOs in a concerted manner, thereby orchestrating their functions. Based on the capacity of Trnp1 to phase separate and its localization surrounding the nuclear MLOs, in particular its enrichment around the nucleoli, we propose a model with Trnp1 levels easing the transition between the MLOs and the surrounding nucleoplasm as depicted in Fig 9.

### Structure–function analysis of Trnp1

Trnp1 has a central structured region flanked by two IDRs at the N- and C-terminus. IDR proteins have been described to act as "hubs" in protein interaction networks (Dunker *et al*, 2001; Darling &

Uversky, 2018) assembling specific proteins together in MLOs (Banani *et al*, 2017) by a LLPS mechanism (Damianov *et al*, 2016; Feric *et al*, 2016; Strom *et al*, 2017). To do so, IDR proteins have to initiate low-affinity transient interactions with many other proteins and also form homo-oligomers simultaneously (Li *et al*, 2012; Banani *et al*, 2017). Trnp1 does indeed both—interact with many proteins associated with MLOs and with itself. Our structure–function analysis showed that Trnp1 IDRs are important for both types of interactions. Both IDRs are needed for the homo-oligomerization capacity of Trnp1 and also exert effects on its capacity to phase separate. Interestingly, the highly conserved first 16 N-terminal amino acids of Trnp1 are most critical for LLPS since droplet size is strongly reduced when this part is deleted. Moreover, most of the interactions of Trnp1 with other proteins are lost upon deletion of this stretch of amino acids. Accordingly, deletion of these N-terminal 16 aa also abrogates the effects of Trnp1 on nucleolar and heterochromatin architecture as well as its function in speeding up M-phase entry and cell cycle progression *in vitro* and *in vivo*. Interestingly, *in vivo* this Trnp1 truncation even exhibits some degree of dominant-negative effects. While Trnp1 increases the proportion of NSCs, the expression of Δ1–16Trnp1 reduces it, while increasing TAP proliferation. Likewise the effects of Trnp1 on increasing nucleolus size were rather the opposite with Δ1–16Trnp1 reducing size and number of nucleoli and heterochromatin spots. Therefore, this small region of the N-terminal IDR exerts powerful effects in virtually all functional aspects of Trnp1 explored here.

Although this small region is composed by LC amino acids as part of the N-terminal IDR, it is 100% conserved among all Trnp1 orthologs in line with its importance for Trnp1 function. It has been shown that post-translational modifications (PTMs) play an important role in the conservation of IDRs during evolution (Narasumani & Harrison, 2018) and their capacity to phase separate (Owen & Shewmaker, 2019). As several of these N-terminal aa of Trnp1 can be post-translationally modified, mostly by phosphorylation and oxidation, such modifications may be also crucial for the fast regulation of Trnp1 function. It is also well conceivable that deletion of these 16aa affects the secondary and tertiary structure of Trnp1. This may also explain the partial dominant-negative effects observed by the expression of this deletion construct mostly *in vivo* in the presence of endogenous Trnp1. In this regard, it is also important to note that the Δ1–16Trnp1 can still interact with WT Trnp1 and may thereby exert dominant-negative effects only *in vivo* in the

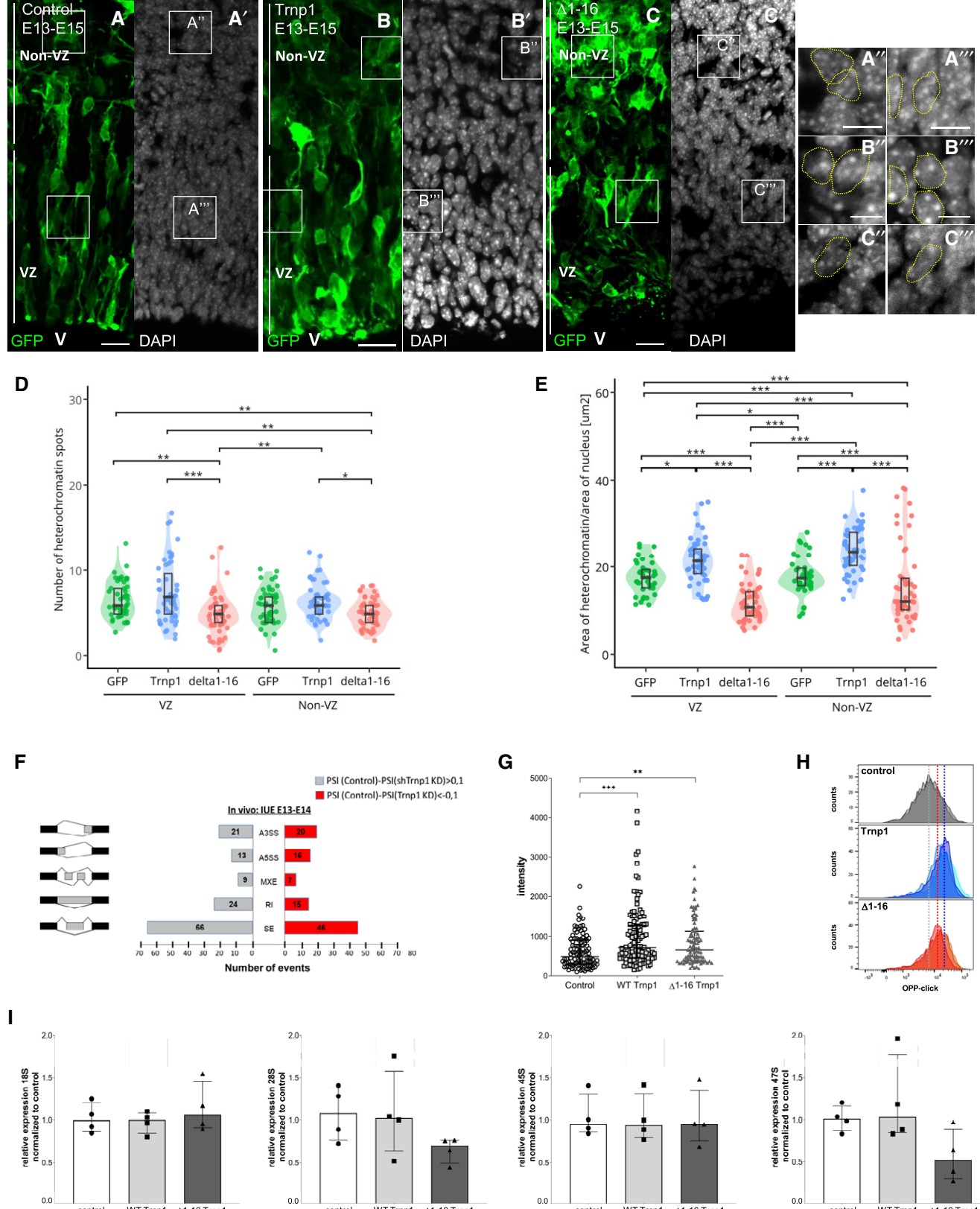

Figure 8.

**Figure 8.  Trnp1 co-regulates the function of several nuclear MLOs.**

A–C    Micrographs of sections of E15 cerebral cortex 2 days after IUE with plasmids and magnifications indicated. V = ventricle; VZ: ventricular zone. Scale bars: 20 and 10 µm for magnifications. Yellow dotted shapes show examples of heterochromatin spots in nuclei of electroporated cells from the VZ and the non-VZ.

D, E    Violin plots depicting the number (D) and size (E) of condensed chromatin spots from cells electroporated as in (A–C). Note that Trnp1 overexpression is sufficient to significantly increase the size of condensed chromatin spots, while the Δ1–16Trnp1 acts as a dominant negative reducing the number and size of heterochromatin spots. N = 10 cells per region from 5 embryos for GFP and Trnp1 and 8–15 cells from 4 embryos for Δ1–16Trnp1 of at least 4 litters. Each dot represents a cell. Results are shown as median ± IQR.

F    Summary of the splicing changes observed in GFP+ cells isolated by FAC sorting from E14 cerebral cortex 24 h after IUE with plasmids expressing the shRNA against Trnp1 and its control as determined by RNA-seq analysis. The number of events per category is shown in the bar. SE, skipped exon; RI, intron retention; MXE, mutually exclusive exon; A5SS, alternative donor site; A3SS, alternative acceptor site.

G, H    Translation assay using OPP-click reaction in P19 cells transfected for 24 h with GFP, Trnp1-IRES-GFP, or Δ1–16Trnp1-IRES-GFP plasmids and analyzed by immunofluorescence (G) or FAC sorting (H). Graph in (G) is representative from three different biological replicates. Each dot represents a cell. Control = 102; Trnp1 = 126; Δ1–16Trnp1 = 97. Graph in (H) shows analysis of 2,000 transfected cells from 1 representative experiment with 2 biological replicates for control and 3 biological replicates for Trnp1 and Δ1–16Trnp1 conditions. The same FAC sorting experiment was repeated two more times. The dotted vertical line shows the mean of the OPP click signal for each condition.

I    Bar and dot plots depicting rRNA transcription for the 18S, 28S, 45S, and 47S measured by qPCR in P19 cells transfected for 24 h with plasmids expressing Trnp1-IRES-GFP or Δ1–16Trnp1-IRES-GFP compared to control cells expressing GFP. Each dot represents a biological replicate. N = 4 for each condition.

Data information: Kruskal–Wallis test with Dunn's *post hoc* test. P-values *P < 0.05; **P < 0.01; ***P < 0.005.

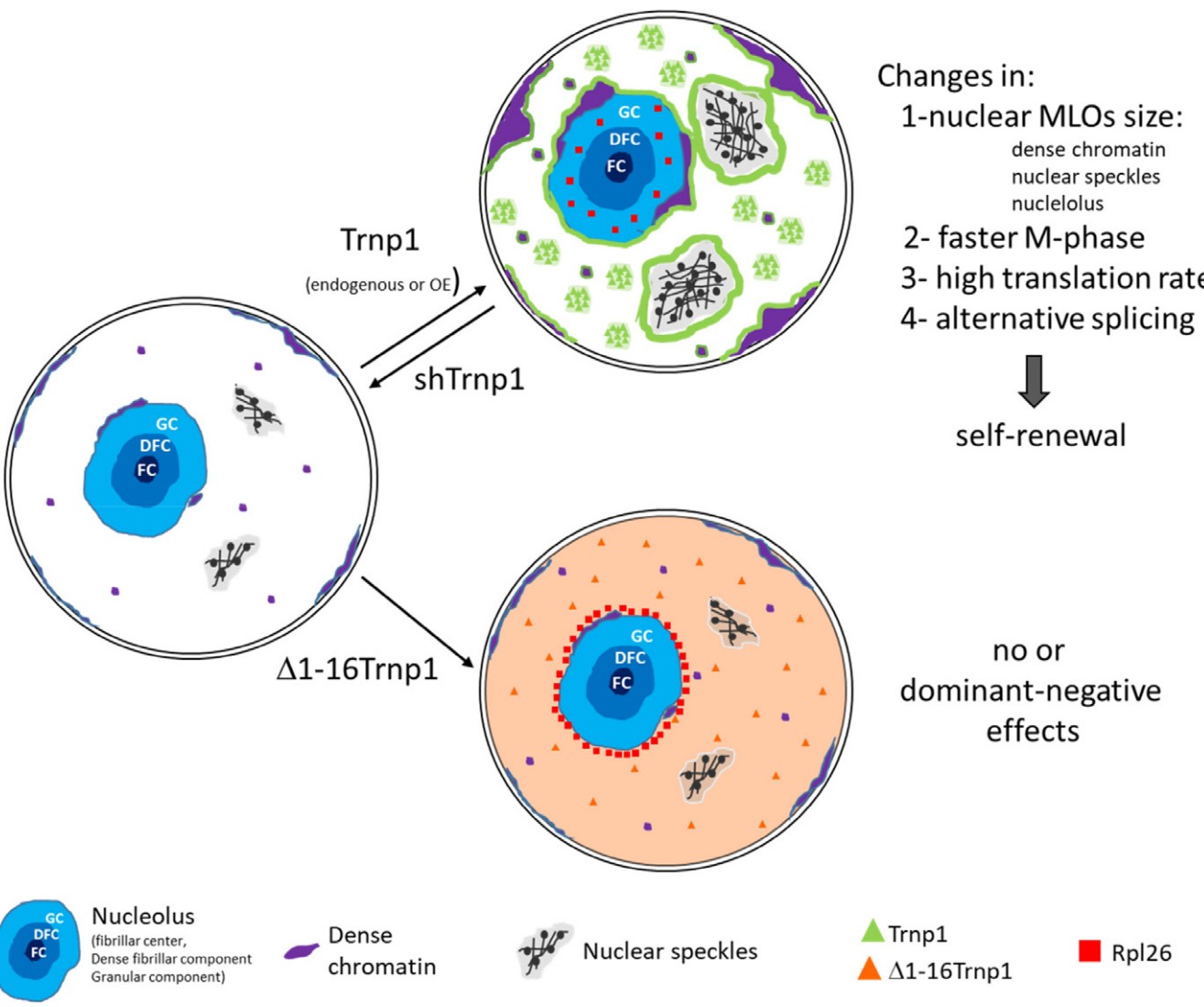

**Figure 9.  Model of Trnp1 function.**

See manuscript text for details.

presence of endogenous Trnp1, but not in P19 cells in the absence of endogenous Trnp1.

Beyond its IDRs, Trnp1 has a structured alpha helix region, containing a CC domain, which also promotes heterotypic interactions as many of Trnp1 interactors are also proteins with a CC domain. In fact, Trnp1 was first identified as an interactor of the CC containing E3 ubiquitin ligase TMF/ARA160 (Volpe *et al*, 2006), and accordingly, we found that this conserved part is important to regulate Trnp1 protein levels via proteasome. Trnp1 protein levels are key in the development of embryonic cortex, with even slight differences exerting profound effects on expansion of stem and progenitor cell types needed for brain folding (Stahl *et al*, 2013; Martinez-Martinez *et al*, 2016). Thus, our data add a new mode to regulate Trnp1 levels to the previously reported transcriptional regulation (Stahl *et al*, 2013; de Juan Romero *et al*, 2015).

## Trnp1 regulates nucleolar architecture and function and M-phase length

A particularly important nuclear MLO accumulating proteins of entire pathways and including specific genes inside its boundaries is the nucleolus. Our proteome analysis showed that Trnp1 interacts with many nucleolar proteins with important roles for its function, biogenesis, and cell cycle progression. Furthermore, we could demonstrate that Trnp1 regulates nucleolar dynamics, architecture, and translation, supporting the idea that Trnp1 is a new regulator of this nuclear MLO. During interphase, the different steps for ribosome biogenesis are responsible for the three sub-compartments (GC, DFC, and FC) nucleolar organization (Tchelidze *et al*, 2019). We found that Trnp1 concentrates in the surrounding of the nucleolus overlapping with the GC domain-containing Ncl, one of the most abundant nucleolar proteins and one of the Trnp1 interactors. This localization and the fact that Trnp1 interacts with many other nucleolar proteins, including ribosomal proteins transported from the cytoplasm (Thomson *et al*, 2013), inspire the hypothesis that Trnp1 could be involved in the regulation of the protein traffic between nucleoplasm and nucleoli. Indeed, staining for the ribosomal protein Rpl26 showed its accumulation surrounding the nucleoli upon expression of the Trnp1 constructs with deletion of the N-terminal IDR (Fig EV3B and C). Our MS analysis shows loss of most interaction partners after deletion of the N-terminal 16 aa, suggesting that this may hamper the transport into the nucleolus. We would therefore suggest that Trnp1 could control ribosome biogenesis and the rate of translation by interacting with nucleolar proteins and thereby ease the protein traffic into the nucleolus.

We showed here by overexpression and knock-down experiments *in vivo* that higher Trnp1 levels increase and lower levels decrease the size of nucleoli in the developing cerebral cortex. Moreover, increasing Trnp1 levels also increases the rate of translation, in agreement with previous work showing that the size of the nucleolus is positively correlated with the ribosome production (Derenzini *et al*, 1998). The function of Trnp1 in transport of ribosomal proteins into the nucleolus and hence ribosomal biogenesis could also be at least partly responsible for the effects of Trnp1 in promoting proliferation as increased production of ribosomes has also been correlated with increased proliferation rates (Lempiainen & Shore, 2009).

Like several other nuclear MLOs, nucleoli disassemble as cells enter mitosis (Dundr *et al*, 2000) and then reassemble at the end of mitosis to ensure survival of the daughter cells. Using continuous live imaging, we observed that Trnp1 increased the speed of nucleoli disassembly but not re-assembly, an effect that is correlated with a faster mitosis. Although the cells expressing Trnp1 displayed a normal progression along the different phases of mitosis, we found that the faster mitosis progression was predominantly due to faster entry of cells into M-phase and a shorter prophase, when nucleoli are disassembled. In contrast, telophase was not affected consistent with Trnp1 not regulating the speed of nucleolus re-assembly at the end of mitosis. These data indicate that Trnp1 co-regulates at the same time the speed of the initial phases of mitosis and the disassembly of nucleoli.

Trnp1 could do that by different means. One important function of the nucleolus is to sequester key effector molecules away from their site of action (Rai *et al*, 2018). Several of these effectors are proteins related to progression of cell cycle such as enzymes responsible of PTM of cell cycle proteins (Boisvert *et al*, 2007). Trnp1, by regulating the nuclear-nucleolus traffic of these proteins, could regulate their liberation from the nucleoli promoting a faster entry of cells into mitosis and, as consequence, a faster disassembly of nucleoli. Thus, by regulating nucleolar protein trafficking Trnp1 could also regulate cell cycle progression and nucleolus disassembly. Alternatively, or in addition, Trnp1 may also regulate these processes by regulating the time that chromosomes need to condensate, an essential step for nucleolus disassembly (Boisvert *et al*, 2007). Indeed, we could show here that Trnp1 interacts with proteins essential for chromosome condensation such as histones; AKAP8; mitotic kinases like Plk1; and the Condensin complex proteins Ncapd2, Ncapg, and Ncaph. The nuclear protein AKAP8 is necessary to recruit HDAC3 at the onset of mitosis and promote the de-acetylation of histone H3 which brings the Condensin complex to the chromatin (Han *et al*, 2015). Plk1, localized in the chromosomes axes, promotes the phosphorylation of the Condensin complex, a process necessary for the correct chromosomes condensation in prophase (Abe *et al*, 2011). Thus, by regulating the function and/or localization of these proteins Trnp1 could control the speed of chromosome condensation and as consequence the speed of nucleoli disassembly.

These effects of Trnp1 on M-phase length are particularly relevant in NSC differentiation, since prolongation of M-phase has been shown to promote differentiation of NSCs in the developing cortex (Pilaz *et al*, 2016). This suggests that shortening of M-phase would result in reduced NSC differentiation and increased self-renewal, as observed when Trnp1 levels are increased, here and in (Stahl *et al*, 2013).

## Trnp1 affects several nuclear MLOs

Trnp1 also affects the architecture of heterochromatin spots and nuclear speckles, other well-described nuclear MLOs that are also disassembled during M-phase (Spector & Lamond, 2011), confirming the importance of Trnp1 function in proliferating cells. This observation is further supported by our MS data showing Trnp1 to interact with many proteins responsible for chromatin condensation and heterochromatin formation, such as all the proteins of the NuRD, PRC2 and PRC1 complex (Laugesen & Helin, 2014; Basta & Rauchman, 2015), and several proteins localized in the nuclear speckles. Since Trnp1 is not part of these

complexes but, like for nucleoli, is localized surrounding these structures (Fig 9), we propose that Trnp1 can also play an important role concentrating and assembling these complexes in specific micro-domains in the nucleus and facilitate their biochemical reactions. In fact, we have shown that Trnp1 regulates gene expression (Stahl *et al*, 2013), alternative splicing and protein translation, confirming the biological importance of Trnp1 for the function of these nuclear MLOs. Trnp1 has thus a rather unique role in co-regulating these MLOs. A recent knock-down screening of MLO components highlighted that few hits regulate 2 or more MLOs (Berchtold *et al*, 2018). These components were all located within the respective MLOs. Trnp1 is rather unique by surrounding the MLOs it affects and thereby regulating their architecture and function (Fig 9).

Taken together, these data support a close link between LLPS properties, interactors, and the functional effects of Trnp1 as an interactor hub in the regulation of nuclear MLOs. Importantly, we show that the effects of Trnp1 on maintaining NSC in a self-renewing state are accompanied by a significant modification in the architecture (nucleoli, heterochromatin) and function (splicing) of different nuclear MLOs also *in vivo*, confirming the biological importance of this new function of Trnp1. As Trnp1 is expressed in many stem cells of other organs during embryogenesis (Metzger, 2011) and various cancer cells (Uhlen *et al*, 2017), its novel role in co-regulating nuclear MLOs and M-phase progression may have a broader relevance beyond the developing nervous system.

# Materials and Methods

### Droplet assay *in vitro*

For droplet formation assay, all the purified proteins were diluted to 100 mM in storing buffer before diluting in droplet buffer (50 mM Sorenson's buffer (pH 7.6) containing 150 mM salt and 2 mM DTT) to the concentrations indicated in the Figures and phase separation was induced by addition of acTEV protease (Invitrogen) at 25°C. RNA or Dextran was used when explicitly mentioned in the Figures. Phase-contrast was used to image the droplets. YFP was used as a negative control for droplet formation. The experiments were repeated three independent times. Note that phase separation properties, i.e., critical concentration for droplet formation, differ slightly between different protein preparations. Quantification of droplets was counted manually using FIJI/ImageJ software.

### Mass spectrometry

For LC-MS/MS purposes, desalted peptides were injected in an Ultimate 3000 RSLCnano system (Thermo) and separated in a 15-cm analytical column (75 µm ID with ReproSil-Pur C18-AQ 2.4 µm from Dr. Maisch) with a 50 min gradient from 5 to 60% acetonitrile in 0.1% formic acid. The effluent from the HPLC was directly electrosprayed into a LTQ-Orbitrap mass spectrometer (Flag-Trnp1 vs. Trnp1 Ref399) or a Q Exactive HF (Flag-delta 1–16_Trnp1 vs. delta 1–16_Trnp1 Ref1644).

For LTQ-Orbitrap measurements, the instrument was operated in data-dependent mode to automatically switch between full scan MS and MS/MS acquisition. Survey full scan MS spectra (from $m/z$ 300 to 2,000) were acquired in the Orbitrap with resolution $R = 60,000$ at $m/z$ 400 (after accumulation to a "target value" of 500,000 in the linear ion trap). The six most intense peptide ions with charge states between 2 and 4 were sequentially isolated to a target value of 10,000 and fragmented in the linear ion trap by collision-induced dissociation (CID). All fragment ion spectra were recorded in the LTQ part of the instrument. For all measurements with the Orbitrap detector, three lock-mass ions from ambient air were used for internal calibration. Typical MS conditions were as follows: spray voltage, 1.5 kV; no sheath and auxiliary gas flow; heated capillary temperature, 200°C; normalized CID energy 35%; activation $q = 0.25$; and activation time = 30 ms.

For Q Exactive HF measurements, the mass spectrometer was operated in data-dependent mode to automatically switch between full scan MS and MS/MS acquisition. Survey full scan MS spectra (from $m/z$ 375 to 1,600) were acquired with resolution $R = 60,000$ at $m/z$ 400 (AGC target of $3 × 106$). The 10 most intense peptide ions with charge states between 2 and 5 were sequentially isolated to a target value of $1 × 105$ and fragmented at 27% normalized collision energy. Typical mass spectrometric conditions were as follows: spray voltage, 1.5 kV; no sheath and auxiliary gas flow; heated capillary temperature, 250°C; and ion selection threshold, 33,000 counts.

MaxQuant 1.5.2.8 was used to identify proteins and quantify by iBAQ with the following parameters: Database, Uniprot_Mmusculus_UP000000589_10090_151030; MS tol, 10 ppm; MS/MS tol, 0.5 Da (Orbitrap LTQ) or 10 ppm (QExactive); Peptide FDR, 0.1; Protein FDR, 0.01; Min. peptide Length, 5; Variable modifications, Oxidation (M); Fixed modifications, Carbamidomethyl (C); Peptides for protein quantitation, razor, and unique; Min. peptides, 1; Min. ratio count. 2. Identified proteins were considered as interaction partners, if their MaxQuant iBAQ values displayed a greater than 1.9-fold change enrichment and *P*-value 0.05 (ANOVA) when compared to the control.

Statistical and bioinformatics analysis was performed with the Perseus software (Tyanova *et al*, 2016) (version 1.5.6.0), Microsoft Excel, and R statistical software. Proteins that were identified in the decoy reverse database or only by site modification were not considered for data analysis. We also excluded potential contaminants. Data were further filtered to make sure that identified proteins showed expression in all biological triplicates and missing values were imputed on the basis of normal distribution. Mean $\log_2$ ratios of biological triplicates and the corresponding p values were visualized with volcano plots. We used *t*-test for binary comparisons and SAM with FC > 1.9 and *P* < 0.05 for the assessment of *t*-test results in volcano plot. The DAVID webserver (Version 6.8, LHRI & DAVID Bioinformatics) was employed for Gene Ontology enrichment analysis and KEGG pathway analysis (Huang da *et al*, 2009). Trnp1 interactors were also subjected to STRING analysis using the web-tool available via http://string-db.org/ (Jensen *et al*, 2009). Figure 4D shows the confidence view with the active prediction methods "Experiments", "Databases", and highest confidence score (0.9) hiding disconnected nodes. A Kmeans clutering of 4 was applied.

### Time-lapse imaging

P19 cells were co transfected with either pCAG-RFP, or pCAG-Trnp1-IRES-RFP or pCAG-Δ1–16 Trnp1-IRES-RFP and the reporter

plasmids GFP-Fbl1 or H2B-Venus, respectively. Cells were plated on PDL-coated glass bottom microwell dishes (MatTek) with a glass thickness of 0.16–0.19 mm. Imaging was performed on an inverse confocal laser scanning microscope (FV1000, Olympus) using a 20× air objective with NA = 0.7 (UPLAPO). During imaging, the cells were incubated in a humidified stage-top incubator (Tokai Hit) at 37°C with 5% $CO_2$. Confocal image stacks were acquired every 10 min with a z-step size of 1.5 μm for a total of eight optical sections. The total imaging time was 16–72 h. For quantification, the z-stacks were collapsed using the maximum projection in FIJI/ImageJ.

To analyze the time a cell spent in different sub-phases of M-phase, we identified the movie-frame when a cell first showed condensation of the nuclear DNA by following H2B-Venus and then measured the time until a clear separation of the two sister chromatids was detectable (condensation to separation) and subsequently measured the time until decondensation of the nuclear DNA (separation to end of M-phase). Total length of M-phase was determined by the time from first condensation of nuclear DNA to the end of DNA condensation.

In order to quantify the number and size of nucleoli around M-phase, we employed multiple functions of FIJI/ImageJ. First, the cell of interested was isolated by removing all fluorescence signals originating from other cells. To quantify the number and the area of GFP-Fbl1 foci, we normalized the signal from each cell by employing the "Enhance Contrast" function aiming at 0.3% saturated pixels in all movies. In order to get a better quantifiable binary picture, we then used the "Threshold" function with default values followed by "Watershed". To then assess the number and size of GFP-Fbl1 foci, we used the "Analyze Particle" function. The quantitative data derived from the movies in FIJI/ImageJ were then imported to R and centered around the first picture-frame when the separation of the daughter cells was detectable.

### Quantitative analysis of in utero electroporated cells

To quantify the distribution and percentage of Pax6, Tbr2, Ki67, BrdU, and/or pHH3 electroporated cells, single optical sections Olympus FV1000 confocal laser-scanning microscope) of each brain slice were analyzed. Using FIJI/ImageJ, each stripe was divided into five equally sized bins, adjusted to the radial thickness of the embryonic cerebral cortex. Quantifications are given as median ± IQR. Different sections representing the full rostro-caudal distribution of the electroporated area were used for quantifications of each condition. For each condition (overexpression, knockdown and corresponding controls), 4–8 embryos from 2 to 5 litters were analyzed.

Number and volume measurements of nucleoli and condensed chromatin spots in vivo were performed on Z-stack images acquired with Olympus FV1000 confocal laser-scanning microscope using a 63×/1.35 N.A. water immersion objective. Whole nuclei were recorded with a z-step size of 1 μm. Nucleoli, condensed chromatin spots, and nuclei were counted and assigned to cells manually. Values were saved and transferred to FIJI/ImageJ. The measurements were performed for Ncl staining (present in DFC and GC) and DAPI to determine nucleolar and condensed chromatin spots and nuclei volumes, respectively. The resulting values were transferred to an Excel sheet, and we used R software to calculate the average

area and the total number per cell. The results were displayed as violin plots or dot plots, on which the data points represent individual cells. For each condition (overexpression and corresponding controls), 10–15 cells were quantified 6 (GFP), 6 (for Trnp1), 3–4 (for Δ1–16Trnp1), 7 (shTrnp1), and 7 (shcontrol) different embryos (from 3 to 5 different litters) for nucleoli and 6 (GFP and Trnp1) and 4 (Δ1–16Trnp1) different embryos (from 3 to 5 different litters) for condensed chromatin.

### Animals

Animal handling and experimental procedures were performed in accordance with German and European Union guidelines and were approved by the State of upper Bavaria. All efforts were made to minimize suffering and number of animals. Two to three month female C57BL/6J wild-type mice were maintained in specific pathogen-free conditions in the animal facility, in 12:12-h light/dark cycles and bred under standard housing conditions in the animal facility of the Helmholtz Center Munich and the Biomedical Center Munich. The day of the vaginal plug was considered embryonic day (E) 0.

### Plasmids

The pEGFP-C1-Fibrillarin (26673) and pCMV-L26-Flag (19972) were purchased in Addgene. The murine Trnp1-expressing plasmid was previously described (Stahl et al, 2013). Empty vectors were used as controls. For sequence of primers used, see Table EV2. All plasmids for expression were cloned from pENTR1a-Trnp1 gateway plasmid described in Stahl et al (2013) into a Gateway (Invitrogen) form of pCAG-GFP and pCAG-dsRed (kind gift of Paolo Malatesta). For that purpose, we first used PCR to amplify the truncated forms from the murine Trnp1-expressing plasmid using the different primers containing specific restriction sites described in Table EV2. Cloning was done by restriction digestion. Gateway LR-reaction system was used to then sub-clone the different Trnp1 forms into the pCAG destination vectors.

To produce the different recombinant 6HisxMBP-tagged Trnp1 proteins used in this study, we cloned all the sequences from the pCAG plasmids into the plasmid pET-MBP1a (kind gift of Arie Geerlof) using PCR amplification with specific primers containing specific restriction sites. Automated sequencing was used to check all constructs.

### Cell lines

P19 cells (ATCC, VA, USA, CRL-1825) were maintained in DMEM GlutaMAX™ (Life Technologies) supplemented with 10% FCS, 1× NEAA (Life Technologies) and 1% penicillin/streptomycin (Life Technologies).

### Primary cortex cultures

E14 cortices were dissected removing the ganglionic eminence, the olfactory bulb, the hippocampal anlage, and the meninges, and cells were mechanically dissociated with a fire polish Pasteur pipette. Cells were then seeded onto poly-D-lysine-coated glass coverslips in DMEM-GlutaMAX with 10% FCS (Life Technologies).

## Plasmid transfection

Plasmid transfection was done with Lipofectamine 2000 (Life technologies) according to manufacturer's instruction, and 24 h later, cells were washed in PBS collected for protein extraction or fixed in 4% paraformaldehyde (PFA) in buffered in phosphate-buffered saline (PBS) and processed for immunostaining.

## M-phase analysis

P19 cells transfected for 16 h with plasmids expressing GFP (control) or Trnp1 with GFP were synchronization at end G2 by treatment for 24 h with 9 mM of RO-3306 (Calbiochem). Cells were then released for the different time points depicted in Fig 6F, fixed with 4% PFA, and stained for GFP and DAPI to visualize the chromatin. Pictures for all conditions were done using epifluorescence microscope (Zeiss, Axio ImagerM2) in a 20×/0.85 N.A water immersion objective. Hundred cells were manually counted in each condition to calculate the percentage of cells in each mitotic phase (interphase, prophase, metaphase, anaphase, and telophase).

## Super-resolution confocal microscopy (HyVolution)

Super-resolution confocal image was performed with an inverted Leica SP8X WLL microscope, equipped with a pulsed WLL2 laser (470–670 nm) and acusto-optical beam splitter. 0.8 μm thick image stacks at a z-step size of 200 nm were acquired with a 100×/1.4 NA oil immersion objective, scan speed was 100 Hz, and image pixel size was set to 39 nm. The pinhole was set to 0.6 AU (calculated for 633 nm). The following fluorescence settings were used: B23, Alexa Fluor 488: (excitation: 488 nm; emission: 496–535 nm), Ncl, Alexa Fluor 546: (561; 569–600), and Trnp1, Abberior STAR 635P: (633; 643–701). Recording was done line sequentially to avoid bleed-through. A triple notch filter (NF 488/561/633) was used to block reflected excitation light. All signals were recorded with hybrid photodetectors (HyDs) in counting mode. Stacks were deconvolved using Huygens Professional 17.10 p2. Images shown are single planes of deconvolved stacks.

## Transmission electron microscopy

Trnp1-IRES-GFP or GFP-transfected P19 cells were FACS sorted for GFP at 1 day after transfection using a FACS Aria II (BD) in BD FACS Flow TM medium. Debris and aggregated cells were gated out by forward scatter (FSC-A) and side scatter (SSC-A); FSC-A vs. FSC-W was used to discriminate doublets from single cells. Gating for GFP+ cells was done using non-transfected cells. The FACS-sorted cells were then washed in PBS (pH 7.2) and fixed in 1.6% glutaraldehyde diluted in 0.1 M PBS (pH 7.2) at room temperature (RT); after 1 h, they were washed in the same buffer. The samples were acetylated according to Thiry *et al* (1985) before being embedded in Epon. Ultrathin sections mounted on copper grids were stained with uranyl acetate and lead citrate before examination with a Jeol JEM 1400 transmission electron microscope (TEM) at 80 kV.

## Immunostaining

Tissue sections (20 μm) or cells plated on poly-D-lysine-coated glass coverslips were blocked with 2% BSA, 0.5% Triton-X (in PBS) for

1 h prior to staining. Primary antibodies (see Table EV1) were applied in blocking solution overnight at 4°C. Fluorescent secondary antibodies were applied in blocking solution for 1 h at room temperature. DAPI (4′,6-diamidin-2-phenylindol, Sigma) was used to visualize nuclei. Sections were mounted in Aqua Polymount (Polysciences). All secondary antibodies were purchased from Life Technologies. Images were taken using an Olympus FV1000 confocal laser-scanning microscope using 20×/0.85 N.A and 63×/1.35 N.A. water immersion objective or epifluorescence microscope (Zeiss, Axio ImagerM2) equipped with a 20×/0.8 N.A and 63×/1.25 N.A. oil immersion objectives. Post-image processing with regard to brightness and contrast was carried out where appropriate to improve visualization, in a pairwise manner. See Table EV1 for the list of primary and secondary antibodies. For staining of Trnp1, a pretreatment with 2 N HCl for 30 min and subsequently neutralized with sodium-tetraborate ($Na_2B_4O_7$ 0.1 M, pH: 8.5) for 2 × 15 min was done.

## Analysis of protein co-localization by line scan

Protein co-localization after staining was determined using Olympus FV1000 confocal laser-scanning microscope pictures and FIJI/ImageJ software. Single z-stack images were taken at the foci. A region of interest line (shown in yellow line) was drawn simultaneously in all channels analyzed so that it spanned the nucleus and intersected the foci (Fig 4). Data of relative staining intensity vs. distance were exported and processed using Microsoft Excel.

## Analysis of protein turnover

P19 cells were transfected with the appropriate expression construct. Twenty hours after transfection, cycloheximide (Sigma-Aldrich) was added to the cultures to a final concentration of 20 μM to inhibit new protein synthesis and protein lysates were collected at different time points. To measure protein turnover after proteasome inhibition, the cultures were treated in an identical manner except that 10 μM MG132 was added to the cultures. Equal amounts of protein in lysates from the different time points were then analyzed by Western blot (WB). ImageJ was used to quantify protein content in each sample and to calculate the rate of the different protein degradation by normalizing to the GFP protein (half-life of 24 h) of the same sample.

## Protein extract preparation

P19 cells were transfected with the appropriate expression plasmid for 24 h. To obtain nuclear extracts, cells were first washed in ice-cold 1× PBS, re-suspended in Tween20 lysis buffer (25 mM HEPES pH 8, 20 mM NaCl, 2 mM EDTA, 1 mM PMSF, 0.5% Tween20, 1× protease inhibitors, cOmplete, Roche), and incubated on ice for 30 min with gentle vortexing every 10 min. Nuclei were then pelleted by centrifugation at 336 × *g* 10 min at 4°C, and the cytoplasmic fraction (supernatant) was then discarded or collected for the nuclear cytoplasmic fractionation experiment. The nuclear fraction was washed once with lysis buffer, re-suspended lysis buffer containing 500 mM NaCl, incubated 30 min on ice, and sonicated 30 times for 1 sec ON/1 sec OFF at 30% input. Chromatin and cell debris were pelleted at 13,500 × *g* for 15 min at 4°C, and the

supernatant containing the nuclear proteins was recovered in a low-protein-binding Eppendorf. For mass spectrometry experiments, DNAseI (250 U; SIGMA) was added to the nuclear pellet and incubated 1 h at 37°C. Finally, after a centrifugation at $16,000 \times g$ for 15 min at 4°C, the supernatant was collected, mixed with the nuclear proteins and the protein lysates (cytoplasmic and nuclear depending on the experiment were snap-frozen to use in different experiments).

For whole protein extract, cells were incubated 30 min with RIPA buffer (Sigma) completed with $1\times$ protease inhibitors, cOmplete (Roche) prior to sonicate 30 times for 1 ON/1 sec OFF at 30% input. Chromatin and cell debris were pelleted at $13,500 \times g$ for 15 min at 4°C, and the supernatant containing the nuclear proteins was recovered in a low-protein-binding Eppendorf and snap-frozen for later experiments.

## Protein co-immunoprecipitation (IP)

For co-immunoprecipitation, the nuclear protein extract was incubated end-over-end with 10 μg of primary antibodies or FLAG® M2 Magnetic Beads (SIGMA) at 4°C for 2 h. When using antibodies, 30 μl of a 1:1 mix of Dynabeads® Protein G: Dynabeads® Protein A (Invitrogen) was used and incubated end-over-end at 4°C for 1 h. Beads were then separated with a magnet and washed four times for 15 min at 4°C with washing buffer ((25 mM HEPES pH 7.6, 150 mM KCl, 12.5 mM MgCl$_2$, 0.1 mM EDTA, 10% glycerol) + 0.1% NP-40). For WB analysis, proteins were eluted by cooking beads at 95°C with mixer at 600 rpm for 10 min in 2× Laemmli buffer and separated from the beads with a magnet. For LC-MS/MS, beads were subsequently washed three times with 50 mM NH$_4$HCO$_3$, incubated with 10 ng/μl trypsin in 1 M urea 50 mM NH$_4$HCO$_3$ for 30 min, washed with 50 mM NH$_4$HCO$_3$ again, and digested ON in the presence of 1 mM DTT. Digested peptides were alkylated and desalted prior to LC-MS analysis.

## Western blot

Gel electrophoresis was done with 10 or 15% SDS-polyacrilamide gels depending of the molecular size of the respective protein and then transferred to nitrocellulose membranes. For immunodetection, membranes were first blocked with 5% nonfat dry milk in TBS/T (Tris-buffered saline/0.1%Tween-20, pH 7.4) overnight, incubated for 3 h with primary antibodies diluted in 1% nonfat dry milk in TBS/T at RT and, after extensive washing with TBS/T, incubated with horseradish peroxidase (HRP)-coupled secondary antibodies diluted in 1% nonfat dry milk in TBS/T for 1 h. Signal was visualized by ECL method on films (Amersham). For antibodies, see Table EV1.

## RNA extraction, cDNA synthesis, and qPCR

For qPCR analysis, P19 cells were transfected with the appropriate expression constructs for 24 h and total RNA was isolated using the Qiagen RNeasy kit. cDNA synthesis was performed using random primers with the Maxima first strand synthesis kit (Thermo Scientific). Real-time qPCR was conducted using SYBR green and Thermo Fisher Quant Studio 6 machine. PCR primers are listed Table EV2.

For droplet experiments, RNA from E14 embryonic cortices was isolated using the Qiagen RNeasy kit, quantified using Qubit® RNA HS Assay Kits (Life Technologies) according to the manufacturer, and snap-frozen in small aliquots until use.

## Recombinant protein expression and purification

All His$_6$-MBP-tagged Trnp1 constructs and His$_6$-MBP-tagged YFP were transformed into *Escherichia coli* strain BL21 (DE3) and cultured at 20°C in 2-l flasks containing 500 ml ZYM 5052 auto-induction medium (Studier, 2005) and 100 μg/ml kanamycin. Cells were harvested by centrifugation after reaching saturation, resuspended in 30 ml lysis buffer (50 mM Tris–HCl, 1 M NaCl, 20 mM imidazole, 10 mM MgCl$_2$, 10 mg/ml DNaseI, 1 mM AEBSF.HCl, 0.2% (*v/v*) NP-40, 1 mg/ml lysozyme, 0.01% (*v/v*) 1-thioglycerol, pH 8.0), and lysed by sonication. The lysates were clarified by centrifugation ($48,000 \times g$) and filtration (0.2 μM). The supernatants of His$_6$-MBP-tagged Trnp1, Δ1−16, Δ1–140, and Δ95–223 were subjected to polyethylenimine (PEI) precipitation to remove non-specifically bound nucleotides. 0.5% PEI (in 0.5 M Tris–HCl pH 8.0) was added to the supernatants and incubated for 10 min at 4°C while stirring vigorously. The precipitate was removed by centrifugation ($48,000 \times g$). To remove the excess of PEI, ammonium sulfate was added (95% saturation) to the supernatants and incubated for 30 min at 4°C while stirring vigorously. Precipitated proteins were collected by centrifugation ($75,000 \times g$), and the pellets were redissolved in 30 ml buffer A (50 mM Tris–HCl, 300 mM NaCl, 20 mM imidazole, 0.01% (*v/v*) 1-thioglycerol, pH 8.0), centrifuged to remove insoluble material, and applied to 5-ml HiTrap Chelating HP column (GE Healthcare), equilibrated in buffer A using an Äkta Purifier (GE Healthcare). The column was washed with buffer A containing 50 mM imidazole, and bound proteins were eluted with buffer B (50 mM Tris–HCl, 300 mM NaCl, 300 mM imidazole, 0.01% (*v/v*) 1-thioglycerol, pH 8.0). Fractions containing protein were subsequently subjected to size exclusion chromatography using a HiLoad 16/600 Superdex 200 column (GE Healthcare) and equilibrated in buffer C (50 mM Tris–HCl, 300 mM NaCl, and 0.01% (*v/v*) 1-thioglycerol, pH 8.0). Fractions containing Trnp1 constructs were pooled and concentrated to 10–30 mg/ml, flash-frozen in small aliquots, and stored at −80°C.

His$_6$-MBP-tagged Trnp1CC and Trnp1mutCC and His$_6$-MBP-tagged YFP were purified as described above without the PEI and ammonium sulfate precipitation steps. For the purification of His$_6$-MBP-tagged Trnp1CC and Trnp1mutCC, the IMAC buffers contained 1 M NaCl.

Static light scattering (SLS) experiments of MBP-tagged Trnp1CC and Trnp1mutCC were performed at 30°C using a Viscotek TDA 305 Triple Array Detector (Malvern Instruments) downstream to an Äkta Purifier (GE Healthcare) equipped with an analytical size exclusion column (Superose 6 10/300 GL, GE Healthcare) at 4°C. 100 μl samples were run at a flow rate of 0.5 ml/min in 50 mM Tris–HCl, 300 mM NaCl, 0.01% (*v/v*) 1-thioglycerol, pH 8.0, with a concentration of 1.91 mg/ml for CC and 1.19 mg/ml for mutCC, respectively. His$_6$-MBP-tagged Trnp1-CC was pre-purified on the superpose 6 column to remove highly aggregated protein, which would have given a very strong scattering signal that would have interfered with the analysis of the SLS experiments. The molecular masses of the samples were calculated from the refractive index and right-angle light-scattering signals using Omnisec (Malvern Instruments). The

SLS detector was calibrated with a 4 mg/ml BSA solution with 66.4 kDa for the BSA monomer and a d$n$/d$c$ value of 0.185 ml/g for all protein samples.

### *In utero* electroporation

Animals were operated as approved by the Government of Upper Bavaria. E13 timed pregnant mice were anaesthetized by intraperitoneal (i.p.) injection of Fentanyl (0.05 mg/kg), Medetomidine (0.5 mg/kg) and Midazolam (5 mg/kg). Plasmids were mixed with 0.1% Fast Green (Sigma) and injected at a concentration of 1 µg/µl. *In utero* electroporations were performed as described earlier (Saito, 2006). Anesthesia was terminated by subcutaneous injection of Buprenorphine (0.1 mg/kg), Flumazenil (0.5 mg/kg) and Atipamezole (2.5 mg/kg), and the mice were allowed to recover on a heating pad and monitored according to the license. The embryonic brains were collected 1 or 2 days after electroporation. To be able to analyze the number of cells in S-phase, Bromodeoxyuridine-5-bromo-2′-deoxyuridine (BrdU) was injected I.P. one hour prior to sacrifice the animals.

### Tissue processing

Mouse E14 and E15 embryonic mouse brains were fixed for 4 and 5 h, respectively, in 4% PFA in PBS at 4°C. Brains were then cryoprotected in 30% sucrose (in PBS) overnight, embedded in tissue-tek, stored at −20°C, and then cryosectioned.

### FACS analysis

For the *in vivo* experiment, E13 developing cortices were electroporated with plasmid expressing the shRNA against Trnp1 together with GFP or its control plasmid. Electroporated cerebral cortices were collected 1 dpe (at E14) for FACS analysis. Three separate biological replicates were performed with each replicate containing 10,000 GFP+ cells. For that, electroporated E14 cerebral cortices were enzymatically dissociated with 0.5% Trypsin, at 37°C for 15 min. After dissociation, samples were washed in PBS by centrifugation at 300 $g$ for 10 min. In both experiments, cell suspension was filtered through a 100-µm cell strainer and placed on ice for analysis. FACS analysis was performed at a FACSAria III (BD Biosciences) in FACSFlow sheath fluid (BD Biosciences), with a nozzle diameter of 100 µm. Debris and aggregated cells were gated out by forward and side scatter; single cells were selected by FSC-W/FSC-A. Gating for GFP fluorescence was done using non-electroporated/transfected cells. Flow rate during sorting was below 500 events/s.

### RNA sequencing

RNA from FACS sorted *in utero* electroporated (GFP+) cells expressing the shRNA against Trnp1 together with GFP or its control was isolated in Extraction Buffer (Arcturus), heated to 42°C, and stored at −80°C until all samples were collected to be processed together. Subsequently, total RNA was isolated using the PicoPure RNA Isolation Kit as per the manufacturer's instructions (Arcturus). cDNA was synthesized from 300 pg of total RNA using SMART-Seq v4 Ultra Low Input RNA Kit for Sequencing (Clontech), according to the manufacturer's instructions. Prior to generating the final library

for Illumina sequencing, the Covaris AFA system was used to perform the cDNA shearing, resulting in 200- to 500-bp-long cDNA fragments. The quality and concentration of the sheared cDNA were assessed on Agilent 2100 Bioanalyzer before proceeding to library preparation using MicroPlex Library Preparation kit v2 (Diagenode). Final libraries were evaluated and quantified using an Agilent 2100 Bioanalyzer, and the concentration was measured additionally with Quant-iT PicoGreen dsDNA Assay Kit (ThermoFisher) before sequencing. The uniquely barcoded libraries were multiplexed onto one lane, and 150-bp paired-end deep sequencing was carried out on HiSeq 4000 (Illumina) that generated ~ 30 million reads per sample.

The RNA-seq reads were mapped to the mouse genome reference (UCSC genome browser mm10) using mapsplice (Wang *et al*, 2010) with default parameters. The percent splice in (PSI) values were calculated for each alternative splicing events using the MISO pipeline (Katz *et al*, 2010). To identify Trnp1-related splicing events, we used the cutoffs: delta PSI > 0.1 or < −0.1 with Bayes factor > 5 and $P$ value < 0.1(Student's $t$-test) between Trnp1 overexpression and GFP control overexpression for the *in vitro* analysis and Trnp1 knockdown and its control in the *in vivo* analysis. Gene expression levels were estimated using RSEM (Li & Dewey, 2011). Differentially expressed genes were identified using the R package "DESeq2", and the cutoff is | log$_2$FoldChange| > 1 and $P$ value < 0.05.

### OPP-click translation assay

Translation rate was measured using the Protein Synthesis Assay Kit (Cayman chemicals, 601100) in cells expressing RFP as control or Trnp1 and RFP for 24 h. Following manufacturer's instructions, for OPP quantifications in immunostaining, all images within compared groups were acquired with identical settings and analyzed in FIJI/ImageJ software by quantification of pixel intensity. Quantifications by FACS were performed in the FACSAria III (BD Biosciences) and analyzed using FlowJo 10.6.2.

### Statistical analyses

All manual counts were performed blind. Per experimental condition, quantifications in mouse brain sections were performed with at least four brains from 2 to 4 different litters (see above). Quantifications of primary cortical cells were done with brains from at least three different litters per experimental condition. Quantification of cells in live imaging experiments is derived from three different experiments per experimental condition. Graphs were done with GraphPad Prism 8, designer gravit (https://designer.gravit.io/), or R software. Data were statistically analyzed with GraphPad Prism 8 or R software using Mann–Whitney test or Kruskal–Wallis test with Dunn's *post hoc* test, as indicated throughout the manuscript.

## Data availability

The mass spectrometry proteomics data have been deposited to the ProteomeXchange Consortium via the PRIDE partner repository with the dataset identifier PXD018350 (http://www.ebi.ac.uk/pride/archive/projects/PXD018350).

RNA-seq data of this study are available under accession number PRJNA622767 which can be found in https://www.ncbi.nlm.nih.gov/bioproject/PRJNA622767/.

**Expanded View** for this article is available online.

## Acknowledgements

The authors would like to thank the BMC core facilities for Imaging and Flow Cytometry of the LMU Munich for providing equipment, suggestions and excellent support in the execution of experiments. We are particularly grateful to Dorothee Dormann and Saskia Hutten for plasmids and experimental suggestions in regard to phase transition of Trnp1 and to Michael Sattler for great advice on the biochemistry and structure of Trnp1. We would also like to thank Andrea Steiner-Mezzadri for excellent technical assistance and the members of the Götz lab for discussions. This work was funded by the advanced ERC grant ChroNeuroRepair (340793) and the German Research Foundation grants in the collaborative research center 870, the priority programs 1757, 1739, and 2202 and the NSFC grant number 31730110.

## Author contributions

The original idea for the study came from MG. MG and ME then initiated the study and planned the experimental approach. ME, SF, and MG designed the experiments; ME performed the molecular, biochemistry, and the *in vivo* and *in vitro* experiments. SF performed some *in vivo* experiments, the time-lapse imaging experiments, and analysis; AM-S performed the quantification of the nucleoli *in vivo* in fixed tissue, and SF and SN analyzed the data. IF and AI performed the mass spectrometry experiments and analysis. SZ and ZW performed the alternative splicing analysis. DN designed the mutantCC Trnp1 protein. AG produced the recombinant proteins. MT performed the EM experiments and analysis. MG and ME wrote the manuscript with input from SF, IF, AG, DN, and AI.

## Conflict of interests

The authors declare that they have no conflict of interest.

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
