## [Review Process File · The EMBO Journal]

Trnp1 organises diverse nuclear membrane-less compartments in neural stem cells

Miriam Esgleas, Sven Falk, Ignasi Forne, Marc Thiry, Sonia Najas, Sirui Zhang, Aina Mas, Arie Geerlof, Dierk Niessing, Zefeng Wang, Axel Imhof, and Magdalena Götz

DOI: N/A

Corresponding author(s): Magdalena Götz (magdalena.goetz@helmholtz-muenchen.de)

Review Timeline:

Submission Date:	6th Sep 19
Editorial Decision:	27th Oct 19
Revision Received:	31st Mar 20
Editorial Decision:	22nd Apr 20
Revision Received:	18th May 20
Accepted:	20th May 20

Editor: Daniel Klimmeck

Transaction Report:

Dear Magdalena,

Thank you for the submission of your manuscript (EMBOJ-2019-103373) to The EMBO Journal. Please accept my sincere apologies for the unusual delay with the peer-review of your manuscript. It was particularly difficult to get referees assigned at this time of the year. Your manuscript has been initially sent to three reviewers, however one reviewer got much delayed and in the end did not send us his-her report after repeated chasers. We have received reports from the other two referees, which I enclose below, and decided to proceed with our decision based on these reports.

As you will see, the referees acknowledge the potential interest and novelty of your results, although they also express a number of major issues that will have to be conclusively addressed before they can be supportive of publication of your manuscript in The EMBO Journal. In more detail, the consistent core areas of the referees' criticism focus around the endogeneous relevance of your results, causalities between Trp1 and the individual phenotypes observed as well as mechanistic depth provided. In addition, the reviewers raise a number of points related to consistency between the results, additional controls required to corroborate the findings, overall data and method representation as well as wording, that would need to be conclusively addressed to achieve the level of robustness and clarity needed for The EMBO Journal.

I judge the comments of the referees to be generally reasonable and given their overall interest, we are in principle happy to invite you to revise your manuscript experimentally to address the referees' comments.

We do concur with the reviewers that providing conclusive support for an endogeneous relevance of your observations will be required to move on with this work for publication in The EMBO Journal.

Note that while per se well taken, more mechanistic insights into Trp1's role in membrane-less organelle organization are not strictly required for the current manuscript on our view.

Please let me know any time if you have additional questions or need further input on the referee comments.

Please see below for additional instructions for preparing your revised manuscript.

Thank you for the opportunity to consider your work for publication. I look forward to your revision.

Kind regards,

Daniel

Daniel Klimmeck, PhD
Editor
The EMBO Journal

When assembling figures, please refer to our figure preparation guideline in order to ensure proper formatting and readability in print as well as on screen:
<http://bit.ly/EMBOPressFigurePreparationGuideline>

Before submitting your revision, primary datasets (and computer code, where appropriate) produced in this study need to be deposited in an appropriate public database (see <https://www.embopress.org/page/journal/14602075/authorguide#datadeposition>).

The accession numbers and database should be listed in a formal "Data Availability" section (placed after Materials & Method) that follows the model below (see also <https://www.embopress.org/page/journal/14602075/authorguide#availabilityofpublishedmaterial>). Please note that the Data Availability Section is restricted to new primary data that are part of this study.

Data availability

Our journal also encourages inclusion of *data citations in the reference list* to directly cite datasets that were re-used and obtained from public databases. Data citations in the article text are distinct from normal bibliographical citations and should directly link to the database records from which the data can be accessed. In the main text, data citations are formatted as follows: "Data ref: Smith et al, 2001" or "Data ref: NCBI Sequence Read Archive PRJNA342805, 2017". In the Reference list, data citations must be labeled with "[DATASET]". A data reference must provide the database name, accession number/identifiers and a resolvable link to the landing page from which the data can be accessed at the end of the reference. Further instructions are available at <https://www.embopress.org/page/journal/14602075/authorguide#referencesformat>

- a point-by-point response to the referees' comments, with a detailed description of the changes

made (as a word file).

- a word file of the manuscript text.

- individual production quality figure files (one file per figure)

- a complete author checklist, which you can download from our author guidelines (<http://emboj.embopress.org/authorguide>).

- Expanded View files (replacing Supplementary Information)

Further information is available in our Guide For Authors:

The revision must be submitted online within 90 days; please click on the link below to submit the revision online before 25th Jan 2020.

Link Not Available

Referee #2:

The manuscript by Esgleas et al describes the function of trnp1 in regulating the organization of the nucleolus and other membrane-less organelles. The authors first bring insights into the biochemical properties of trnp1. They show that trnp1 is able to self-interact and self-separate, the 16 first aa of the protein being important for trp1 capacity to phase separate, probably through heterotopic interaction with proteins from MLOs. Authors then link this ability to phase separate to the trpn1 function in promoting proliferation in vitro and in vivo in the developing cortex. Overexpression of trnp1 but not trnp1 lacking the 16 first aa, is enhancing neuronal progenitors proliferation. They finally show that trnp1 overexpression (but not delta 1-16) increases the speed of nucleoli assembly that is correlated with a shortening of the duration of mitosis. Finally, the authors show, in vivo, that trnp1 increases the size of nucleoli in region where endogenous trnp1 is lowly express. Although solely based on overexpression, the in vitro experiments provided by Esgleas et al are convincing and demonstrate a novel role of trp1 in regulating MLOs. These results suggest that the regulation of nucleoli architecture might participate in cortical extension in mammalian brain. The authors could have provided more mechanistical insights. Indeed, authors do not fully exploit their interactomic results, that highlight many putative candidates. The impact of trnp1-induced increase of the size of the nucleolus is much less documented in vivo and deserve further investigation to substantiate the conclusion.

Major points

1) Homo-oligomers versus droplets formation : as the authors are not using the same constructs to assess the capacity of trnp1 to self-assemble and to phase separate (2E vs 2F), it is difficult to conclude about the link between both. Do the delta 1-140, delta95-223 and delta1-16 proteins retain the same ability to self-assemble? Could the authors make a correlation between capacity to form Homo-oligomers and capacity to form droplets? Could the authors provide quantification of the droplet size in 2F?

2) Figure 3: To really appreciate the importance of the first 16 aa for proliferation in vivo, authors should provide the control condition (empty vector) and compare the delta 1-16 to the control. As such, one can evaluate if this truncated form totally or partially lose the ability to promote self-renewal. If partial, then other part of the protein might also be involved and would worth testing the effect of overexpression of the Cter protein (see point 3). To strengthen the results, authors should also provide analysis of cell cycle exit as well as neuronal output (% of GFP+ cells that are neurons).

3) Effect of the Cter protein? Why the Cter protein is not considered in most of the manuscript?
a. even if the effect is milder, the droplets formed with the N-term IDR appear smaller than with the WT (2F). Quantification (point 1) might also help evaluating the importance of the Cter part.
b. Fig 2G: authors conclude that the first 16aa at the Nter are key for the trnp1 function in promoting proliferation. From the data, none of the truncated protein (included the one lacking the Cter) promote self-renewal. Why the authors exclude any effect of the Cterminus?

4) Most of the experiments presented in the manuscript are based on overexpression of trnp1 (mostly in a cell type that does not express trnp1 at the endogenous level).
a. Authors show GC localization of overexpressed (5A and 5B) trnp1 in P19 cells. To substantiate their conclusion, authors should provide colocalization (Ncl and trnp1) experiments (super resolution) or Electronic micrographs.
b. The role of the accumulation of trnp1 at the external part of the GC is not clear.
i. Is the endogenous protein accumulating at the GC? Not clear from the images provided in 2C (E14 cortex cells). Authors should provide same line scan analysis that those shown in 5A-B.
ii. Delta 1-18 proteins are still localized at the GC but lack most of the interaction with nucleolar proteins, suggesting that other regions of the protein are important for trnp1 localization at the GC. Have the authors investigate the domain of localization at the GC (maybe using the different truncated variants)? Does the lack of this domain impair the localisation of nucleolar protein (see point iii)?
iii. Is the recruitment of nucleolar protein changing upon overexpression of the delta 1-18 protein? From figure 7D, it seems to be the case, could the authors quantify the distribution of Ncl and other nucleolar proteins (including some ribosomal proteins) in both condition?
iv. From an evolutionary point of view (Stahl et al), it would be interesting to compare localization of endogenous trnp1 in mouse and human stem cells. Would one expect accumulation at the GC or a total increase?

5) The shape of the nucleoli and the nuclei upon trnp1 overexpression (TEM images) does not look as smooth as in the Control condition, could the authors comment on that? Could trnp1 OE have a function in maintaining these shapes?

6) Dynamic of nucleoli during cell cycle. 6F: Would have been more informative to analyze all the mitotic phases independently (prophase, metaphase, anaphase). To test if the mitosis length is increased upon overexpression of trnp1 or trnp1 delta 1-18 in vivo, authors should have analysis the mitotic index (pHH3 staining) one day after in utero electroporation.

7) Figure 8 :

a. authors have chosen to analyse number and size of the nucleoli in two regions VZ (RGCs - APs) and non-VZ (TAPs, IPs). However these two regions may encompass different cell types, especially comparing control and *trnp1* OE conditions: I would recommend authors to present their results differently and assess the number and size of nucleoli in the different cell population (costaining Pax6+, *tbr2*+, *tbr1*+, NeuN).

b. Further investigations in E14 cortex cells will really benefit to the study. In fact, authors do not provide any evidences for a link between nucleolar architecture and cell cycle progression in vivo the developing cortex. Authors could at least investigate this in vitro on culture of neuronal stem cells.

c. On the same line, is the nucleolus disassembly affected in neuronal progenitors?

d. Authors have previously shown that overexpression of *trnp1* lead to an increase in *pax6* and a decrease in *tbr2*+ cells. If *trnp1* OE promotes self-renewal, why would it not be the case in the IPs? Would worth assessing proliferation status of TBR2+ cells upon *trnp1* OE.

8) As authors are proposing a limitation of *trnp1* in regulating nucleoli size (line 366) and that *trnp1* acts as a novel regulator of nuclear MLOs. To draw those conclusions, authors should have assess nucleoli status in condition depleted for *trnp1* (siRNAs).

Minor points

9) Line 95 : Although, authors bring biochemical data, do they really provide any insight into the structure of *Trnp1*? I would recommend removing the word structural in sentence line 95

10) Localization of *Trnp1* truncated forms: Authors claim that "*trnp1* proteins that lack the Cter part were mainly found in the cytoplasm" (line 145-146). On the images (2B, S1C), *trnp1* delta Cter constructs seem to be expressed in the nucleus as much as in the cytoplasm (in particular for delta 95-223). On the opposite, the authors suggest the Cter constructs to be exclusively nuclear. On the images, the delta1-140 proteins seem to be partially localized in the cytoplasm. Could the authors comment on that? Biochemical fractionation would be appreciated to quantitatively assess the Cytoplasm versus nucleus localization of the different *trnp1* proteins.

11) Mat and methods: section of primary culture of E14 cortices is missing. No information of methods of transfection is given.

12) Figure S2D and S2E, the n is not mentioned, how many times these experiments have been reproduced?

13) S2E: ColP blot is overexposed: do not allow to appreciate the difference, if any. Authors should also provide quantification.

Referee #3:

Esgleas and colleagues characterize the protein *Trnp1* with regard to its phase separation capability and as novel regulator of several nuclear membraneless compartments. They identify a small conserved region at the amino-terminus of *Trnp1*, which is responsible for *Trnp1* interactions with partner proteins and for its nuclear organizational functions. While the data are overall convincing and good quality, the manuscript is largely descriptive, without providing in-depth mechanistic insights. Another caveat of the manuscript is that it relies heavily on over-expression experiments, with loss-of-function and rescue experiments missing.

Main concerns:

- From a conceptual perspective, it is not really clear (and mechanistically not explained) how a single protein should regulate several nuclear MLOs simultaneously. How are Trnp1 functions regulated in a manner that would reflect the specific functions of the different MLOs it controls?
- Another major concern relates to the over-expression of Trnp1. The majority of experiments rely on ectopic over-expression, which is known to generate artefacts. What are the phenotypes associated with Trnp1 depletion or knockout in cells normally expressing Trnp1? How do those phenotypes relate to the over-expression data? Ideally, knockdown and rescue experiments should be performed to consolidate the main findings, e.g. using wild-type Trnp1 and the delta1-16 mutant.
- Similarly, instead of visualizing nuclear localization and behavior of ectopically expressed Trnp1, it would be much better to tag the endogenous Trnp1 locus with a fluorescent protein. With genome editing tools now available such experiments have become feasible, and are often much more informative than ectopic over-expression experiments.
- The mitotic phenotypes could be due to indirect effects. For instance, how do the authors exclude that Trnp1 over-expression alters DNA replication and/or checkpoint functions during S-phase and G2? Mitotic problems such as lagging chromosomes and chromatin bridges are often a consequence of replication stress and checkpoint defects. Again, such experiments should be complemented by loss-of-function analysis, ideally under conditions in which Trnp1 is depleted specifically during mitosis (e.g. using a degron system).
- Line 116: Why is 25uM a "low" concentration? What is the estimated physiological concentration of Trnp1 in the nucleus?
- Line 118: Which "control proteins" do the authors refer to?
- Line 120: Proteolysis is only one way to disassemble MLOs. Others include PTMs as well as dilution effects, e.g. upon nuclear envelope breakdown in mitosis.
- Figure 4: It would seem more intuitive to add the delta1-16 results directly to this figure.
- Similarly, in Figure 5, it would be important to show the localization of wild-type Trnp1 in comparison to the delta1-16 mutant.

Minor comments:

- Lines 71-74, MLOs may also exclude cellular components. This filtering mechanism, including certain components while excluding others, could be explained a bit better.
- Line 74: What is meant by "disassembled during the cell cycle"? Does this refer to mitosis? Or to dynamic assembly and disassembly during the course of the cell cycle? If mitosis is meant, this should be specified, and it should be pointed out that due to the breakdown of the nuclear envelope many nuclear components are diluted, which most likely explains disassembly of MLOs in mitosis.
- Lines 81-84: Perhaps it would be worth mentioning that also transcription complexes and DNA repair compartments have recently been linked to MLOs with features of LLPS (e.g. Guo et al. Nature 2019, Shrinivas et al. Mol Cell 2019, Bojja et al. Cell 2018, Kilic et al. EMBOJ 2019, Pessina et al. NCB 2019).

- The color channels seem not well aligned in Fig. 1C.
- Fig. 1G seems to contain a vertical image artefact between lanes 2 and 3, presumably coming from image compression.
- The discussion seems quite extensive and could probably be shortened if space is an issue.

Referee #2:

The manuscript by Esgleas et al describes the function of trnp1 in regulating the organization of the nucleolus and other membrane-less organelles. The authors first bring insights into the biochemical properties of trnp1. They show that trnp1 is able to self-interact and self-separate, the 16 first aa of the protein being important for trnp1 capacity to phase separate, probably through heterotopic interaction with proteins from MLOs. Authors then link this ability to phase separate to the trnp1 function in promoting proliferation in vitro and in vivo in the developing cortex. Overexpression of trnp1 but not trnp1 lacking the 16 first aa, is enhancing neuronal progenitors proliferation. They finally show that trnp1 overexpression (but not delta 1-16) increases the speed of nucleoli assembly that is correlated with a shortening of the duration of mitosis. Finally, the authors show, in vivo, that trnp1 increases the size of nucleoli in region where endogenous trnp1 is lowly express. Although solely based on overexpression, the in vitro experiments provided by Esgleas et al are convincing and demonstrate a novel role of trnp1 in regulating MLOs. These results suggest that the regulation of nucleoli architecture might participate in cortical extension in mammalian brain. The authors could have provided more mechanistical insights. Indeed, authors do not fully exploit their interactomic results, that highlight many putative candidates. The impact of trnp1-induced increase of the size of the nucleolus is much less documented in vivo and deserve further investigation to substantiate the conclusion.

Major points

1) Homo-oligomers versus droplets formation : as the authors are not using the same constructs to assess the capacity of trnp1 to self-assemble and to phase separate (2E vs 2F), it is difficult to conclude about the link between both. Do the delta 1-140, delta95-223 and delta1-16 proteins retain the same ability to self-assemble? Could the authors make a correlation between capacity to form Homo-oligomers and capacity to form droplets? Could the authors provide quantification of the droplet size in 2F?

Good point and we have done this. We have produced highly pure new recombinant proteins free of DNA and RNA and repeated the experiments for the revision. We could confirm the previous results with this new batch of proteins and have added new examples and quantifications of the droplet formation assay for the most important constructs to Figures 1 and 2.

2) Figure 3: To really appreciate the importance of the first 16 aa for proliferation in vivo, authors should provide the control condition (empty vector) and compare the delta 1-16 to the control. As such, one can evaluate if this truncated form totally or partially lose the ability to promote self-renewal. If partial, then other part of the protein might also be involved and would worth testing the effect of overexpression of the Cter protein (see point 3). To strengthen the results, authors should also

provide analysis of cell cycle exit as well as neuronal output (% of GFP+ cells that are neurons).

We have added the in utero electroporation data for the control construct to Figure 3 and Figure 7. These data show that the loss of these first 16 amino acids interferes with the function of Trnp1 in promoting NSC fate (Figure 3) and nucleoli size increase (Figure 7). Most strikingly, deletion of these first 16 amino acids of Trnp1 does not cause a partial loss-of-function, but even a mild dominant-negative effect indicating that it interferes or competes with the function of endogenous Trnp1.

We already showed in our previous work (Stahl et al., 2013) that Trnp1 promotes an increase of NSCs at the expense of Transit-amplifying progenitors and neurons in vivo compared to the cells expressing the control vector. Here we could confirm these results and in addition we demonstrated in Fig 3D that OE of the delta1-16 promotes a significant increase of GFP+ cells in bin4 containing the cortical plate where neurons are located compared to WT Trnp1.

3) Effect of the Cter protein? Why the Cter protein is not considered in most of the manuscript?

The C-terminal Trnp1 is cytoplasmic (Figure 2B, 2C and EV1D, EV3A) and is hence not considered for the effects on nucleoli, heterochromatin and splicing as all these functions are nuclear.

a. even if the effect is milder, the droplets formed with the N-term IDR appear smaller than with the WT (2F). Quantification (point 1) might also help evaluating the importance of the Cter part.

We have added quantifications now shown in Figure 2F of the revised manuscript.

b. Fig 2G: authors conclude that the first 16aa at the Nter are key for the trnp1 function in promoting proliferation. From the data, none of the truncated protein (included the one lacking the Cter) promote self-renewal. Why the authors exclude any effect of the Cterminus?

As mentioned above the C-terminal Trnp1 is cytoplasmic and hence less relevant for the nuclear functions of Trnp1.

4) Most of the experiments presented in the manuscript are based on overexpression of trnp1 (mostly in a cell type that does not express trnp1 at the endogenous level).

This is indeed a very important point. We have now added new data to Figure 7 (panels G-K) showing that the knock-down of Trnp1 in vivo results in reduced size of nucleoli thereby corroborating the function of endogenous Trnp1 in regulating nucleolar size.

a. Authors show GC localization of overexpressed (5A and 5B) trnp1 in P19 cells. To substantiate their conclusion, authors should provide colocalization (Ncl and trnp1) experiments (super resolution) or Electronic micrographs.

We have included super resolution pictures of Trnp1 in Figure 5A of the revised manuscript. These data demonstrate the droplet-like condensates of Trnp1 in the nucleus and its accumulation surrounding the nucleolus.

b. The role of the accumulation of trnp1 at the external part of the GC is not clear.

i. Is the endogenous protein accumulating at the GC? Not clear from the images provided in 2C (E14 cortex cells). Authors should provide same line scan analysis that those shown in 5A-B.

We included new confocal high magnification images in Figure 1C, 5A and Fig EV3A of the revised manuscript where Trnp1 localisation accumulating around non Trnp1 stained areas (nucleoli among others) is more clear. In addition, Figure 5A and Fig EV3A now shows endogenous Trnp1 and a line scan demonstrating its accumulation surrounding the nucleoli.

ii. Delta 1-18 proteins are still localized at the GC but lack most of the interaction with nucleolar proteins, suggesting that other regions of the protein are important for trnp1 localization at the GC. Have the authors investigate the domain of localization at the GC (maybe using the different truncated variants)? Does the lack of this domain impair the localisation of nucleolar protein (see point iii)?

We think that this domain comprises the 1-16 amino acids of the N-terminal IDR, as the construct lacking this stretch no longer interacts with the nucleolar (and other) proteins (Figure 4E). In addition, we now also provide line scans of the localization of the different deletion constructs of Trnp1 together with a nucleolin staining in Figure EV3A. This shows that the enrichment of Trnp1 surrounding the nucleoli is partially lost when the first 16 amino acids are deleted. Interestingly, the N-terminus truncated Trnp1 proteins partially co-localizes with nucleolin, i.e. in the nucleoli, while C-term deletions then result in more cytoplasmic localization of Trnp1. Finally, we show in Figure EV3B,C that expression of the Trnp1 lacking these 16 N-terminal amino acids results in mislocalization of the ribosomal protein Rpl26, prompting the hypothesis of Trnp1 accumulation in the surrounding of the nucleoli helping the protein transport into the nucleoli.

iii. Is the recruitment of nucleolar protein changing upon overexpression of the delta 1-18 protein? From figure 7D, it seems to be the case, could the authors quantify the distribution of Ncl and other nucleolar proteins (including some ribosomal proteins) in both condition?

As mentioned above, we provide a representative line scan of the colocalization of Trnp1 lacking the first 16 amino acids together with nucleolin and find that the enrichment surrounding the nucleoli is partially lost resulting in more wide-spread distribution also within the nucleoli. Most interesting are the effects of the different Trnp1 truncated constructs on the distribution of the ribosomal protein Rpl26, showing that N-term truncated Trnp1 proteins, including delta 1-16Trnp1, trap most Rpl26 in the surrounding of the nucleoli while no effects on the localization of this protein can be observed in the C-term truncated Trnp1 proteins.

iv. From an evolutionary point of view (Stahl et al), it would be interesting to

compare localization of endogenous trnp1 in mouse and human stem cells. Would one expect accumulation at the GC or a total increase?

We made several attempts to generate an antibody recognizing also the human Trnp1, even in different hosts (mouse, rat and guinea pig), but we failed. In addition, we tested several commercial antibodies (including the companies ABCAM (ab174303), Biorbyt (orb51746 and Orb186386) and Boster Biological Technology (a15615) without success.

5) The shape of the nucleoli and the nuclei upon trnp1 overexpression (TEM images) does not look as smooth as in the Control condition, could the authors comment on that? Could trnp1 OE have a function in maintaining these shapes?

Even after careful re-examination, we found no systematic effect on the shape of the nucleoli in our TEM.

6) Dynamic of nucleoli during cell cycle. 6F: Would have been more informative to analyze all the mitotic phases independently (prophase, metaphase, anaphase). To test if the mitosis length is increased upon overexpression of trnp1 or trnp1 delta 1-18 in vivo, authors should have analysis the mitotic index (pHH3 staining) one day after in utero electroporation.

We have done both of these experiments and added the analysis of the different mitotic phase as Figure 6F and the mitotic index as Figure 6G in the revised manuscript. These data demonstrate that mostly exit of interphase and length of prophase are affected by Trnp1 overexpression and that overexpression of Trnp1 also affects the mitotic index of NSCs in vivo, reducing the time spent in mitosis (Figure 6G). We also further completed the proliferation analysis in vivo by additional data in Figure 3G, adding new data on the proliferation of basal progenitors.

7) Figure 8 :

a. authors have chosen to analyse number and size of the nucleoli in two regions VZ (RGCs - APs) and non-VZ (TAPs, IPs). However these two regions may encompasses different cell types, especially comparing control and trnp1 OE conditions: I would recommend authors to present their results differently and assess the number and size of nucleoli in the different cell population (costaining Pax6+, tbr2+, tbr1+, NeuN).

Indeed, we did almost what the reviewer suggests as we took the morphology of the cells in the different regions into account in our previous analysis – i.e. in the VZ we examined nucleoli and heterochromatin in cells with RGC morphology and in the SVZ with a multipolar basal progenitor morphology. We have now added Figure EV4B-C illustrating that indeed the VZ cells that we analyze are Pax6+ radial glia and the SVZ cells Tbr2+ basal progenitors.

In addition, we also took in account reviewer comment and included Pax6 staining in the new analysis of nucleoli in vivo after Trnp1 knockdown (Fig. 7).

b. Further investigations in E14 cortex cells will really benefit to the study. In fact, authors do not provide any evidences for a link between nucleolar architecture and cell cycle progression in vivo the developing cortex. Authors could at least investigate this in vitro on culture of neuronal stem cells.

To answer this comment, we performed mitotic index analysis in NSCs in vivo after electroporation of *Trnp1*, *delta1-16* and the control (Figure 6G) and we show that mitotic index results correlates with the nucleoli architecture also in vivo (faster mitosis in NSCs with bigger nucleoli).

c. On the same line, is the nucleolus disassembly affected in neuronal progenitors?

Unfortunately, this is not feasible as primary cortex progenitors are heterogeneous. About 50% of Pax6+ RGCs have high levels of *Trnp1*, 50% have low levels, and *Tbr2+* progenitors have no *Trnp1*. In addition, this proportion changes after some days in culture. Thus, following individual cells by time lapse without knowing if they express endogenous *Trnp1* or not will not be conclusive as we do not know if we see no effect of *Trnp1* knock-down as the cell had no *Trnp1* expression already before KD *Trnp1*.

d. Authors have previously shown that overexpression of *trnp1* lead to an increase in *pax6* and a decrease in *tbr2+* cells. If *trnp1* OE promotes self-renewal, why would it not be the case in the IPs? Would worth assessing proliferation status of TBR2+ cells upon *trnp1* OE.

We have added these data as Figure 3G of the revised manuscript. Interestingly WT *Trnp1* does not affect proliferation of *Tbr2+* cells, demonstrating its specificity for NSCs in the cortex. Intriguingly, however, the *delta1-16* that acts like a dominant negative in the NSCs promotes proliferation of the *Tbr2+* cells, also in this regard acting in an antagonistic manner.

8) As authors are proposing a limitation of *trnp1* in regulating nucleoli size (line 366) and that *trnp1* acts as a novel regulator of nuclear MLOs. To draw those conclusions, authors should have assess nucleoli status in condition depleted for *trnp1* (siRNAs).

We have added these new data as Figure 7G-K of the revised manuscript. These experiments show a significant reduction in nucleoli size in the VZ upon *Trnp1* knock-down using the shRNA against *Trnp1* previously verified (Stahl et al., 2013) corroborating the observation that cells expressing endogenous *Trnp1* in vivo have bigger nucleoli (Figure 7F, VZ versus Non-VZ; Fig EV4A). This is particularly notable as only the self-renewing NSCs have high levels of endogenous *Trnp1*, while the differentiating VZ cells, the nascent basal progenitors, have lower levels – yet we see a significant effect supporting the key role of endogenous *Trnp1* in regulating nucleoli size.

Minor points

9) Line 95 : Although, authors bring biochemical data, do they really provide any insight into the structure of Trnp1? I would recommend removing the word structural in sentence line 95

We have removed the word 'structural' there.

10) Localization of Trnp1 truncated forms: Authors claim that "trnp1 proteins that lack the Cter part were mainly found in the cytoplasm" (line 145-146). On the images (2B, S1C), trnp1 delta Cter constructs seem to be expressed in the nucleus as much as in the cytoplasm (in particular for delta 95-223). On the opposite, the authors suggest the Cter constructs to be exclusively nuclear. On the images, the delta1-140 proteins seem to be partially localized in the cytoplasm. Could the authors comment on that?

We now provide line scans of all the truncation constructs to clearly demonstrate their localization in Figure EV3. The staining intensity for the C-terminal deletion constructs (delta 97-233 and delta140-223 is higher in the cytoplasm than in the nucleus, but some signal is present in the nucleus. The coiled-coil mutation is both nuclear and cytoplasmic, while all other constructs are mostly in the nucleus with different localization around or in the nucleolus.

Biochemical fractionation would be appreciated to quantitatively assess the Cytoplasm versus nucleus localization of the different trnp1 proteins.

We have added these data and show the different deletion constructs in the nuclear or cytoplasmic fraction as Figure 2C of the revised manuscript. Together with the line scans provided in Figure EV3A, these data demonstrate clearly that Trnp1 is virtually exclusively nuclear, while the coiled-coil mutation and N-terminal deletions are both in the nucleus and cytoplasm and the c-terminal deletions are predominantly in the cytoplasm.

11) Mat and methods: section of primary culture of E14 cortices is missing. No information of methods of transfection is given.

Thanks for noting this. We have added this description to the Methods section.

12) Figure S2D and S2E, the n is not mentioned, how many times these experiments have been reproduced?

We have added this information to the legend of Figure S2. As stated also in the method section all experiments were repeated at least 3 times.

13) S2E: CoIP blot is overexposed: do not allow to appreciate the difference, if any. Authors should also provide quantification.

We repeated the experiment and added a new representative blot including the single transfected controls to demonstrate this more clearly (Figure 1 and Fig EV2).

Referee #3:

Esgleas and colleagues characterize the protein Trnp1 with regard to its phase separation capability and as novel regulator of several nuclear membraneless compartments. They identify a small conserved region at the amino-terminus of Trnp1, which is responsible for Trnp1 interactions with partner proteins and for its nuclear organizational functions. While the data are overall convincing and good quality, the manuscript is largely descriptive, without providing in-depth mechanistic insights. Another caveat of the manuscript is that it relies heavily on over-expression experiments, with loss-of-function and rescue experiments missing.

Main concerns:

- From a conceptual perspective, it is not really clear (and mechanistically not explained) how a single protein should regulate several nuclear MLOs simultaneously. How are Trnp1 functions regulated in a manner that would reflect the specific functions of the different MLOs it controls?

Our model is that it actually co-regulates the function of these MLOs, which is necessary e.g. in stem cells – fast proliferation, increased translation, splicing and silencing of specific genes. We spell this out more clearly in the discussion, but most importantly added data showing new data on the regulation of splicing (Figure 8E) and translation (Figure 8F-G). Thereby we show clear effects of Trnp1 for all the MLOs with which Trnp1 interacts according to our proteomic analysis functional data. This is why we propose the concept that Trnp1 is for the first time a coordinator of several MLOs simultaneously.

Trnp1 levels are regulated in line with this as they are higher in self-renewing stem cells and this is regulated at transcriptional and post-transcriptional levels.

- Another major concern relates to the over-expression of Trnp1. The majority of experiments rely on ectopic over-expression, which is known to generate artefacts. What are the phenotypes associated with Trnp1 depletion or knockout in cells normally expressing Trnp1? How do those phenotypes relate to the over-expression data? Ideally, knockdown and rescue experiments should be performed to consolidate the main findings, e.g. using wild-type Trnp1 and the delta1-16 mutant.

This is an important point and we have added a new Figure 7 with the new results of nucleoli size reduction in the ventricular zone after in utero electroporation of shRNA against Trnp1. Thus, while Trnp1 overexpression increases the nucleolar size, knock-down of Trnp1 reduces the size of nucleoli, thereby demonstrating a role of endogenous Trnp1 in regulating size, but not number of nucleoli.

Rescue experiments and further validation of these constructs were already published in Stahl et al., 2013.

- Similarly, instead of visualizing nuclear localization and behavior of ectopically expressed Trnp1, it would be much better to tag the endogenous Trnp1 locus with a

fluorescent protein. With genome editing tools now available such experiments have become feasible, and are often much more informative than ectopic over-expression experiments.

Unfortunately, fluorescently tagged Trnp1 acts as a dominant negative as shown in our previous work (Stahl et al., 2013). However, we are in the process of making a mouse line with flag-tagged Trnp1 – but this is clearly beyond the scope of this work. To clarify endogenous nuclear localization of Trnp1 we performed superresolution images of endogenous Trnp1 localisation together with localization of two nucleolar proteins, Ncl and B23 shown in Figure 5A of the revised manuscript.

- The mitotic phenotypes could be due to indirect effects. For instance, how do the authors exclude that Trnp1 over-expression alters DNA replication and/or checkpoint functions during S-phase and G2?

We do not think that Trnp1 overexpression induced check-point activation as our live imaging shows faster division and – as shown in Stahl et al., 2013 and Pilz et al., 2013 – fast symmetric divisions of cells for many cell cycles in vivo and in vitro. To understand better in which phase of the cell cycle Trnp1 is most effective we added a new quantification of all cell cycle phases using synchronized P19 cells as new Figure 6F. These data show that Trnp1 promotes faster exit from interphase and reduces prophase length significantly, but none of the other phases.

Mitotic problems such as lagging chromosomes and chromatin bridges are often a consequence of replication stress and checkpoint defects.

We do not observe chromatin bridges with the exception of a few cells transduced with the delta1-16 construct. As our new data suggest that delta1-16Trnp1 interferes with ribosomal protein entry into the nucleolus, this may indeed interfere with cell cycle progression also indirectly. In vivo, we find more Tbr2+ cells in S-phase (BrdU+) after in utero electroporation of this construct (Figure 3G).

Again, such experiments should be complemented by loss-of-function analysis, ideally under conditions in which Trnp1 is depleted specifically during mitosis (e.g. using a degron system).

This is a very nice suggestion, but in the 3 months of revision (already prolonged by 1 months), we had to prioritize experiments and have prioritized the many in vivo experiments and corroborating the functional effects of Trnp1 on the different MLOs.

- Line 116: Why is 25uM a "low" concentration? What is the estimated physiological concentration of Trnp1 in the nucleus?

We referred to this as low in regard to most experiments with IDR proteins using much higher concentrations. Now we included different protein concentration in the droplet assay shown in Figure 1E and 2F of the revised manuscript.

- Line 118: Which "control proteins" do the authors refer to?

The recombinant MBP-YFP protein. This information is now included in the figures, results section and the Figure legends.

- Line 120: Proteolysis is only one way to disassemble MLOs. Others include PTMs as well as dilution effects, e.g. upon nuclear envelope breakdown in mitosis.

This is a good point and now added to the manuscript (line 121-122).

- Figure 4: It would seem more intuitive to add the delta1-16 results directly to this figure.

We fully agree with the reviewer and have done so as Figure 4E of the revised manuscript.

- Similarly, in Figure 5, it would be important to show the localization of wild-type Trnp1 in comparison to the delta1-16 mutant.

We fully agree and have done so as well as providing high resolution localization of endogenous Trnp1. In the revised manuscript, we included the latter in the main Figure 5, and show all constructs expressed in P19 cells in Figure EV3A.

Minor comments:

- Lines 71-74, MLOs may also exclude cellular components. This filtering mechanism, including certain components while excluding others, could be explained a bit better.

Very good point and we included this now in the introduction of the revised manuscript (line 72-73).

- Line 74: What is meant by "disassembled during the cell cycle"? Does this refer to mitosis? Or to dynamic assembly and disassembly during the course of the cell cycle? If mitosis is meant, this should be specified, and it should be pointed out that due to the breakdown of the nuclear envelope many nuclear components are diluted, which most likely explains disassembly of MLOs in mitosis.

Yes, we were referring to mitosis and the disappearance of nucleoli in mitosis. We specify this more clearly now and mention this mechanism.

- Lines 81-84: Perhaps it would be worth mentioning that also transcription complexes and DNA repair compartments have recently been linked to MLOs with features of LLPS (e.g. Guo et al. Nature 2019, Shrinivas et al. Mol Cell 2019, Boija et al. Cell 2018, Kilic et al. EMBOJ 2019, Pessina et al. NCB 2019).

Many thanks – this is now included in the introduction (lines 84-85).

- The color channels seem not well aligned in Fig. 1C.

This is corrected now.

- Fig. 1G seems to contain a vertical image artefact between lanes 2 and 3, presumably coming from image compression.

This is due to a defect on the film developing because of the AKFA developing machine.

- The discussion seems quite extensive and could probably be shortened if space is an issue.

We significantly shortened many parts of the discussion to make it more concise in the revised manuscript.

Dear Magdalena,

Thank you for submitting your revised manuscript for consideration by The EMBO Journal. Your amended study was sent back to the referees for re-evaluation, and we have received their comments, which I enclose below. As you will see the referee finds that their concerns have been sufficiently addressed and they are now broadly in favour of publication.

Thus, we are pleased to inform you that your manuscript has been accepted in principle for publication in The EMBO Journal, pending some minor issues, which need to be adjusted at re-submission.

Please consider the remaining referee points to see if you could address their minor issues with additional data or revise the discussion and introduce caveats where appropriate.

Please note that as the protein stability and turnover data (EV1 and 2D) appear sound per se, it is fine in our view to keep these included.

We in addition need you to consider a number of minor points related to formatting and data representation, which are listed below.

Please contact me at any time if you need any help or have further questions.

As you may have noticed, every paper now includes a 'Synopsis', displayed on the html and freely accessible to all readers. The synopsis includes a 'model' figure as well as 2-5 one-short-sentence bullet points that summarize the article. I would appreciate if you could provide the bullet points.

Thank you for giving us the chance to consider your manuscript for The EMBO Journal. I look forward to your final revision.

Again, please contact me at any time if you need any help or have further questions.

Kind regards,

Daniel

Daniel Klimmeck PhD
Editor
The EMBO Journal.

>> Figure callouts: There are callouts for a Table S1 and Table S2. do they refer to the EV tables? Please check and correct as suggested below. All tables and datasets need callouts.

>> Dataset EV legends: Tables EV1 and EV2 would best be uploaded as xlsx files and as "Dataset

EV1" and "Dataset EV2". The other two EV tables would need to be renamed accordingly.

>> There is an appendix file with supplementary M&M, but this should be added to the main manuscript.

>> Please update the data availability section with the code for the RNAseq data.

>> Please consider additional changes and comments from our production team as indicated by attached .doc file and leave changes in track mode.

Please see our instructions to authors

Further information is available in our Guide For Authors:

The revision must be submitted online within 90 days; please click on the link below to submit the revision online before 15th Jul 2020.

Link Not Available

Referee #2:

In their revised manuscript, Esgleas et al answered to most of my initial concerns. In particular, authors now provide much more in vivo evidences for a role of trpn1 in regulating MLOs. In particular, they investigate the consequences of trpn1 knock-down. This is clearly strengthening the manuscript.

Although authors have convincingly addressed most of the concerns, I would have like them to provide more data to answer to my first point. I asked them to assess whether there is any correlation between the ability of trpn1 to self-assemble and to phase separate. However, in the current manuscript, they are still not using the same constructs in each type of experiments (fig 2E versus 2F), though they added the delta 1-16 construct in the 2E figure. This is limiting our understanding of the link between the Homo-oligomerization and the droplets formation. They could have at least justify why not using the same constructs. However, they do provide some quantification of the droplets size that was really appreciated.

I have few additional minor comments:

1. Figure 2: although the immuno (EV1D) and fractionation data (2C) in p19 mostly fit, the localization data in cortex cells (2B) seem a bit different (in particular for delta 95-223 that seems to be expressed in the nucleus as much as in the cytoplasm). I would have appreciated the authors to comment on that cell's discrepancies. Could this be explained by the presence of endogenous trpn1 in cortex cells? Unfortunately, the ability of the delta 95-223 construct to interact with the full length trpn1 has not been tested.
2. Line 136 -137 : "Likewise, we also deleted the C-terminal IDR with (delta95-223) or without (Delta140-223) the predicted alpha helix region." I guess there is a mismatch here, Cter IDR with the helix should be delta 140-223.
3. Figure 3G: I guess author are showing % of tbr2+ cells in S phase (short BrdU pulse). However this not indicated in the manuscript (neither in the text nor in the methods).

Referee #3:

Esgleas et al. present a revised manuscript, in which several of the reviewers' points have been addressed. Overall, the manuscript is still rather descriptive, and I am still missing a clear explanation for what makes the N-terminal IDR of Trpn1, in particular the first 16 amino acids, so special, so that Trpn1 is the first nuclear protein regulating multiple MLOs in a concerted manner, as the authors write. The newly added knockdown results, on the other hand, provide a nice counterpart to the overexpression experiments. The following changes should be considered before publication:

- For Figure 8 on the proposed role of Trpn1 as co-regulator of several nuclear MLOs, wouldn't it be important to include Trpn1 delta1-16 in the overexpression studies? After all, the authors state that "the N-terminal IDR exerts powerful effects in virtually all functional aspects of Trpn1 explored here", and Trpn1 delta1-16 is an integral part of the model in Figure 9 and important for key claims of this study.

- The protein stability data (lines 123-129, Fig. EV1A) do not really add something important to the paper, seem somewhat misplaced, and could be easily left out.
- Similarly, it is not clear why the protein turnover data in Fig. 2D are necessary. In my opinion, also these could be left out.
- Molecular weight markers should be added to Western blot data
- In the text to Fig. 2C it is unclear what is meant by heterotypic and homotypic (e.g. do the authors refer to an interaction between Trnp1 CC and Trnp1 wild-type as heterotypic?). Please specify or rephrase.
- Line 116, please check grammar
- Line 140, mutation "of"
- Lines 287 & 288, please check grammar
- Line 490, please check grammar

Referee #2:

In their revised manuscript, Esgleas et al answered to most of my initial concerns. In particular, authors now provide much more in vivo evidences for a role of trpn1 in regulating MLOs. In particular, they investigate the consequences of trpn1 knock-down. This is clearly strengthening the manuscript.

Although authors have convincingly addressed most of the concerns, I would have like them to provide more data to answer to my first point. I asked them to assess whether there is any correlation between the ability of trpn1 to self-assemble and to phase separate. However, in the current manuscript, they are still not using the same constructs in each type of experiments (fig 2E versus 2F), though they added the delta 1-16 construct in the 2E figure. This is limiting our understanding of the link between the Homo-oligomerization and the droplets formation. They could have at least justify why not using the same constructs.

Good point and we have done this. We included now the oligomerisation assay (Figure 2E) using the same constructs we use for the droplet assay (Trnp1, delta1-16, delta1-140 and delta95-223, Figure 2F) and check their capacity to interact with the WT trnp1 (Figure 2E). In addition we have produced highly pure new recombinant proteins free of nucleic acids and repeated the droplet formation in vitro experiments for the revision three times (Figure 1E and 2F). With both experiments we can show that the mutants losing the capacity to interact with the WT Trnp1 (Figure 2E) also form smaller droplets compared to the WT Trnp1 (Figure 2F). We also show that although the delta1-16 is still able to interact with the WT Trnp1 (Figure 2E) it forms smaller droplets compared to the WT Trnp1 (Figure 2F and G). These data suggest that both the N and C-terminal IDRs are important for optimal droplet formation, while only larger deletions in the N-terminal IDR (larger than delta1-16) show deficits in the interaction with WT Trnp1.

However, they do provide some quantification of the droplets size that was really appreciated.

I have few additional minor comments:

1. Figure 2: although the immuno (EV1D) and fractionation data (2C) in p19 mostly fit, the localization data in cortex cells (2B) seem a bit different (in particular for delta 95-223 that seems to be expressed in the nucleus as much as in the cytoplasm). I would have appreciated the authors to comment on that cell's discrepancies. Could this be explained by the presence of endogenous trpn1 in cortex cells? Unfortunately, the ability of the delta 95-223 construct to interact with the full length trpn1 has not been tested.

We apologize for the slightly misleading example and now included another example in Figure 2C showing more clearly now that the delta95-223Trnp1 is mostly cytoplasmic also in primary cortical cells. In addition, we include the self-interaction data for this mutant with the WTTrnp1 in Figure 2E showing that the loss of the Cterm IDR results in a loss of interaction with the WTTrnp1 protein.

2. Line 136 -137 : "Likewise, we also deleted the C-terminal IDR with (delta95-223) or without (Delta140-223) the predicted alpha helix region." I guess there is a mismatch here, Cter IDR with the helix should be delta 140-223.

Sorry, or wording was misleading. The mutant delta95-223Trnp1 lacks amino acids 95 to 223 including the Cterm IDR and the CC, while the delta140-223 only lacks the C term IDR but still has the coiled-coil domain. We rephrased this sentence to make this clearer (line 135-137).

3. Figure 3G: I guess author are showing % of tbr2+ cells in S phase (short BrdU pulse). However this not indicated in the manuscript (neither in the text nor in the methods).

Very good point. We included now in the text and in the methods (line 214 and 1372-1373)

Referee #3:

Esgleas et al. present a revised manuscript, in which several of the reviewers' points have been addressed. Overall, the manuscript is still rather descriptive, and I am still missing a clear explanation for what makes the N-terminal IDR of Trnp1, in particular the first 16 amino acids, so special, so that Trnp1 is the first nuclear protein regulating multiple MLOs in a concerted manner, as the authors write.

We would of course maintain that the discovery of a protein that modulates splicing, nucleoli size and function and heterochromatin size, and find a region mediating this influence on 3 nuclear compartments simultaneously is functional analysis and not descriptive. However, this is probably semantics, because we fully agree that we do not know the posttranslational modifications and/or specific amino acids mediating the interaction with proteins from all 3 compartments as we have shown that these interactions are lost by MS analysis upon deletion of these 1-16 amino acids. To address this a bit more, we included a bit more discussion about this and our ideas how it could work in the discussion. Importantly, however, the 'first' protein, does not mean the 'only' – there may be more proteins localized in the nucleoplasm that do something similar, possibly by other domains/biochemical structures then....

The newly added knockdown results, on the other hand, provide a nice counterpart to the overexpression experiments. The following changes should be considered before publication:

- For Figure 8 on the proposed role of Trnp1 as co-regulator of several nuclear MLOs, wouldn't it be important to include Trnp1 delta1-16 in the overexpression studies? After all, the authors state that "the N-terminal IDR exerts powerful effects in virtually all functional aspects of Trnp1 explored here", and Trnp1 delta1-16 is an integral part of the model in Figure 9 and important for key claims of this study.

We fully agree with this comment and we included now the data for the delta1-16 mutant in Figure 8A-E (heterochromatin analysis), Figure 8G-H (translation analyzed by immunostaining and FACs) and Figure 8I (ribosomal RNA transcription analysis by qPCR).

- The protein stability data (lines 123-129, Fig. EV1A) do not really add something important to the paper, seem somewhat misplaced, and could be easily left out.

Following the suggestion of the Editor we would maintain it in the extended data of the manuscript.

- Similarly, it is not clear why the protein turnover data in Fig. 2D are necessary. In my opinion, also these could be left out.

Following the suggestion of the Editor we would maintain it in the extended data of the manuscript.

- Molecular weight markers should be added to Western blot data

All molecular weight markers are added as color lines representing the different Mw in all the western blots

- In the text to Fig. 2C it is unclear what is meant by heterotypic and homotypic (e.g. do the authors refer to an interaction between Trnp1 CC and Trnp1 wild-type as heterotypic?). Please specify or rephrase.

We rephrase this sentence (line 160-161)

- Line 116, please check grammar

We corrected the grammar

- Line 140, mutation "of"

Done

- Lines 287 & 288, please check grammar

Done

- Line 490, please check grammar

Done

Dear Magdalena,

Thank you for submitting the revised version of your manuscript. I have now evaluated your amended manuscript and concluded that the remaining minor concerns have been sufficiently addressed.

Thus, I am pleased to inform you that your manuscript has been accepted for publication in the EMBO Journal.

Please note that it is EMBO Journal policy for the transcript of the editorial process (containing referee reports and your response letter) to be published as an online supplement to each paper.

Also in case you might NOT want the transparent process file published at all, you will also need to inform us via email immediately. More information is available here:

http://emboj.embopress.org/about#Transparent_Process

Please note that in order to be able to start the production process, our publisher will need and contact you regarding the following forms:

- PAGE CHARGE AUTHORISATION (For Articles and Resources)

[http://onlinelibrary.wiley.com/journal/10.1002/\(ISSN\)1460-2075/homepage/tej_apc.pdf](http://onlinelibrary.wiley.com/journal/10.1002/(ISSN)1460-2075/homepage/tej_apc.pdf)

- LICENCE TO PUBLISH (for non-Open Access)

Your article cannot be published until the publisher has received the appropriate signed license agreement. Once your article has been received by Wiley for production you will receive an email from Wiley's Author Services system, which will ask you to log in and will present them with the appropriate license for completion.

- LICENCE TO PUBLISH for OPEN ACCESS papers

Authors of accepted peer-reviewed original research articles may choose to pay a fee in order for their published article to be made freely accessible to all online immediately upon publication. The EMBO Open fee is fixed at \$5,200 (+ VAT where applicable).

We offer two licenses for Open Access papers, CC-BY and CC-BY-NC-ND.

For more information on these licenses, please visit: <http://creativecommons.org/licenses/by/3.0/> and http://creativecommons.org/licenses/by-nc-nd/3.0/deed.en_US

- PAYMENT FOR OPEN ACCESS papers

You also need to complete our payment system for Open Access articles. Please follow this link and select EMBO Journal from the drop down list and then complete the payment process:

https://authorservices.wiley.com/bauthor/onlineopen_order.asp

Notably, please be reminded that under the DEAL agreement of German scientific institutions with our publisher Wiley, you could be eligible for free publication of your article in the open access format. Please contact either the administration at your institution or our publishers at Wiley (embojournal@wiley.com) for further questions.

On a different note, I would like to alert you that EMBO Press is currently developing a new format for a video-synopsis of work published with us, which essentially is a short, author-generated film explaining the core findings in hand drawings, and, as we believe, can be very useful to increase visibility of the work.

Please see the following link for a representative example:

http://embopress.org/video_EMBOJ-2014-90147

If you have any questions, please do not hesitate to call or email the Editorial Office.

Kind regards,

Daniel

Daniel Klimmeck, PhD
Editor
The EMBO Journal
EMBO
Postfach 1022-40
Meyerhofstrasse 1
D-69117 Heidelberg
contact@embojournal.org
Submit at: <http://emboj.msubmit.net>

Magdalena Götz
EMBO Journal
EMBOJ-2019-1033373